# Chd1 protects genome integrity at promoters to sustain hypertranscription in embryonic stem cells

Aydan Bulut-Karslioglu [1,2 ✉], Hu Jin [3,4,7], Yun-Kyo Kim[5], Brandon Cho[5], Marcela Guzman-Ayala[1,8], Andrew J. K. Williamson[6,9], Miroslav Hejna[3,4], Maximilian Stötzel[2], Anthony D. Whetton[6], Jun S. Song[3,4] & Miguel Ramalho-Santos [1,5 ✉]

Stem and progenitor cells undergo a global elevation of nascent transcription, or hyper-transcription, during key developmental transitions involving rapid cell proliferation. The chromatin remodeler Chd1 mediates hypertranscription in pluripotent cells but its mechanism of action remains poorly understood. Here we report a novel role for Chd1 in protecting genome integrity at promoter regions by preventing DNA double-stranded break (DSB) accumulation in ES cells. Chd1 interacts with several DNA repair factors including Atm, Parp1, Kap1 and Topoisomerase 2β and its absence leads to an accumulation of DSBs at Chd1-bound Pol II-transcribed genes and rDNA. Genes prone to DNA breaks in *Chd1* KO ES cells are longer genes with GC-rich promoters, a more labile nucleosomal structure and roles in chromatin regulation, transcription and signaling. These results reveal a vulnerability of hypertranscribing stem cells to accumulation of endogenous DNA breaks, with important implications for developmental and cancer biology.

[1]Eli and Edythe Broad Center of Regeneration Medicine and Stem Cell Research, Center for Reproductive Sciences and Diabetes Center, University of California, San Francisco, San Francisco, CA, USA. [2]Max Planck Institute for Molecular Genetics, Berlin, Germany. [3]Carl R. Woese Institute for Genomic Biology, Urbana, IL, USA. [4]Department of Physics, University of Illinois, Urbana-Champaign, Urbana, IL, USA. [5]Lunenfeld-Tanenbaum Research Institute and Department of Molecular Genetics, University of Toronto, Toronto, ON, Canada. [6]Stoller Biomarker Discovery Centre, The University of Manchester, Manchester, UK. [7]Present address: Department of Biomedical Informatics, Harvard Medical School, Boston, MA, USA. [8]Present address: Senti Biosciences, South San Francisco, CA, USA. [9]Present address: Thermo Fisher Scientific, Stafford House, UK. ✉email: Aydan.karslioglu@molgen.mpg.de; mrsantos@lunenfeld.ca

Proliferating stem and progenitor cells are net generators of new cellular biomass and therefore have high biosynthetic demand. One way that stem/progenitor cells cope with this demand is to enter a state of hypertranscription, which involves a global elevation of nascent transcriptional output[1]. Hypertranscription is masked by most transcriptional profiling approaches, but has attracted renewed interest recently. Hypertranscription has been documented to occur and play critical roles in embryonic stem (ES) cells[2], the post-implantation epiblast[3], emergence of definitive hematopoietic stem cells[4], primordial germ cells[5], and neurogenesis[6], and may take place in other settings during development, regeneration, and disease[1,7].

The molecular regulation of hypertranscription, or even how it differs from general transcriptional regulation, remains poorly understood. It is expected that hypertranscription involves a coordinated interplay between activating transcription factors, chromatin remodelers, and RNA Polymerases. Some of the players implicated in promoting hypertranscription are the transcription factors Myc and Yap/Taz, the RNA Polymerase regulator pTEFb and the chromatin remodeler Chd1 (reviewed in Percharde et al.[1]). Chd1 is an ATP-dependent chromatin remodeler that binds specifically to H3K4me3 and is found at sites of active transcription[8,9]. Chd1 removes nucleosomal barriers to transcriptional elongation[10] and is required for the optimal activity of RNA Pol I and II[3]. Loss of Chd1 does not affect transcription per se, but it blunts the ability of stem cells to enter hypertranscription in vitro and in vivo[3]. Despite these recent insights, the molecular function of Chd1 in hypertranscribing cells remains unclear.

Hypertranscription is a dynamic phenomenon that is responsive to extrinsic cues[2,4,5], and may therefore share features with ligand-triggered target gene induction. In hormone-responsive cells, target genes are induced via a mechanism that involves the generation of transient endogenous DNA breaks by Topoisomerase II at promoters[11]. Similar induction of target gene transcription mediated by DNA breaks occurs upon exposure to serum or heat shock[12,13], during zygotic genome activation and in neurogenesis[14,15]. DNA breaks may relieve torsional stress and facilitate DNA unwinding and access of RNA Polymerases[16]. Interestingly, endogenous DNA breaks have recently been shown to occur throughout the genome at the promoters of transcribed genes[17], suggesting that the link between DNA breaks and transcription may be more general. It remains unknown how cells coordinate the occurrence of DNA breaks and their repair with transcription, a coordination that is anticipated to be of particular importance in hypertranscribing pluripotent cells.

In this study, we report that Chd1 interacts with DNA repair factors in undamaged ES cells. Chd1 promotes the chromatin recruitment/retention of these factors and the repair of DSBs at the promoters of active RNA Pol II-transcribed genes and rDNA in ES cells. Our results reveal an unexpected interplay between Chd1 and the DNA repair-associated factors Atm, Kap1, and γH2A.X during the resolution of transcription-associated DSBs in ES cells.

## Results

In order to probe the function of Chd1 in the regulation of hypertranscription, we identified its interacting proteins by immunoprecipitation followed by mass spectrometry (IP-MS) using a *Chd1-Flag* knock-in mouse ES cell line[3]. IPs were performed under two different salt concentrations (150 vs 250 mM) to gauge the interaction strengths (Fig. 1a and Supplementary Fig. 1a–b). In physiological salt concentration (150 mM) we detected 314 proteins, the majority of which were lost in higher salt (Fig. 1a, Supplementary Fig. 1b, and Supplementary Data 1).

Previously described Chd1-interacting proteins such as Ssrp1 and Bptf were recovered at high confidence in physiological salt, therefore we focused further analyses on this dataset (Fig. 1a). As expected, putative Chd1-interacting proteins are enriched for factors involved in chromatin and transcriptional regulation (Fig. 1b and Supplementary Data 2). In addition, there is an unexpected enrichment for DNA repair factors among Chd1 interactors, such as the protein kinase Atm, histone variant H2A.X, MRN complex member Mre11, topoisomerase 2β (Top2β), Xrcc factors (Fig. 1a, c). Another repair-associated protein, Kap1, was retained as a Chd1 interactor at high salt immunoprecipitation (Supplementary Fig. 1b). We confirmed interactions of Chd1 with activated Atm (Atm phospho-S1981), Kap1, Parp1 and Top2β in wild-type ES cells via co-immunoprecipitation (Co-IP) (Fig. 1d and Supplementary Fig. 1c). Interaction of Chd1 with the single-stranded repair factor Xrcc1 was not detected by Co-IP (Fig. 1d). Therefore, we focused on the double-stranded DNA (DSB) repair components for the remainder of this study.

We next assessed how genetic deletion of *Chd1*[3] affects the levels and localization of its DSB repair interactors. Loss of Chd1 leads to reduced global levels of S139 phosphorylated H2A.X (γH2A.X), a mark of DSB repair, despite slight increases in the levels of H2A.X and Atm kinase, which phosphorylates H2A.X (Supplementary Fig. 1d). Immunofluorescence confirmed that Atm and Top2β levels increase upon Chd1 loss, with the unexpected observation that they accumulate in the nucleolus in ES cells (Fig. 2a). To our knowledge, this pattern of accumulation of DNA repair factors at the nucleolus of undamaged cells has not previously been described. We have previously shown that Chd1 binds directly to rDNA and that loss of Chd1 leads to reduced nascent synthesis of rRNA and fragmentation of nucleoli (Guzman-Ayala et al.[3] and Fig. 2a). Chromatin immunoprecipitation-qPCR (ChIP-qPCR) revealed an accumulation of Atm, pKap1 and γH2A.X at rDNA in control cells, and a decrease in pKap1 and γH2A.X despite higher levels of Atm, H2A.X and Kap1 at rDNA in *Chd1* KO ES cells (Fig. 2b and Supplementary Fig. 1e). Taken together, these results suggest that Chd1 modulates the function of interacting factors involved in DSB repair in undamaged ES cells, particularly at rDNA in nucleoli.

We previously used mouse genetics to show that Chd1 is essential for rapid growth of the early post-implantation mouse epiblast (E5.5–6.5) by promoting a high transcriptional output, notably of nascent rRNA at the nucleolus[3]. The surprising presence of γH2A.X in undamaged ES cells and its dependence on Chd1 led us to investigate the status of this histone mark in vivo. We found that γH2A.X is detected in vivo in control embryos at E5.5 and more abundantly at E6.5, and is mainly localized to the nucleolus along with a diffuse nuclear pattern (Fig. 2c, d and Supplementary Fig. 2a). Strikingly, *Chd1* KO embryos entirely lack this γH2A.X signal at both developmental stages (Fig. 2e), while global H2A.X is retained (Supplementary Fig. 2b). These findings in vivo are in agreement with the ES cell data above, showing they are not an artifact of cell culture. The early post-implantation epiblast is one of the fastest proliferating cell types in mammals, with doubling times between 2 and 8 h[18]. Interestingly, the fastest rates of proliferation were recorded at E6.5, where we find high levels of nucleolar γH2A.X (Fig. 2 and Supplementary Fig. 2) Overall, the results indicate that nucleolar accumulation of γH2A.X in vivo correlates with rRNA synthesis and proliferation rate, and all three of these are dependent on Chd1 (this study and Guzman-Ayala et al.[3]).

The loss of γH2A.X in *Chd1* KO cells could simply be a consequence of the reduced global transcriptional output[3] and therefore a lower occurrence of transcription-induced DNA breaks. We therefore set out to determine the levels and genomic location of DSBs in control vs *Chd1* KO ES cells. We performed

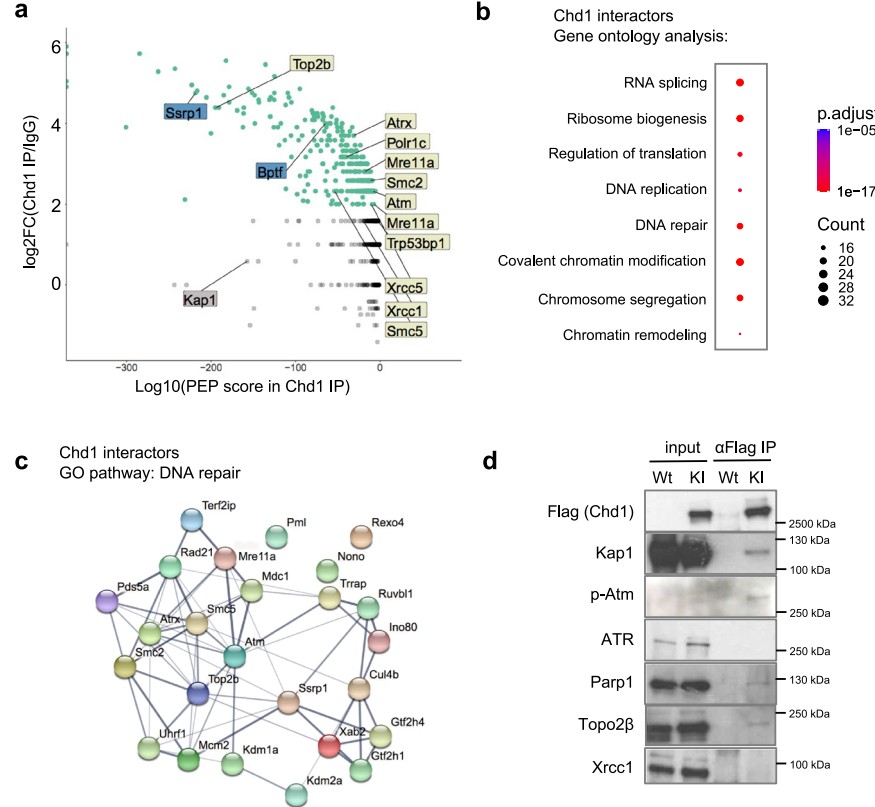

**Fig. 1 Chd1 interacts with double-stranded DNA repair proteins in ES cells. a** Putative Chd1 interactors, identified by IP-mass spectrometry using a *Chd1-Flag* knock-in ES cell line. IgG IP was performed as negative control. Each dot denotes an identified protein. Green dots indicate proteins that are significantly enriched in Chd1 IP vs IgG IP. Gray dots indicate detected, but not significantly enriched, proteins. Previously documented interactors of Chd1 are indicated in blue boxes. Key proteins involved in DNA repair are indicated in cream boxes. Trim28, an interactor identified in the 250 mM salt IP (Supplementary Fig. 1) is indicated in the gray box. The posterior error probability (PEP) score denotes the probability that the identified peptide is correct. False discovery rate was set to 1%. (Full list is available in Supplementary Data 1. **b** Gene ontology analysis of co-immunoprecipitated proteins. Selected pathways are shown. The color scale shows the adjusted *p* value of the enriched pathways. Full list is available in Supplementary Data 2. **c** Protein interaction network of factors co-immunoprecipitated with Chd1 and belonging to the gene ontology term "DNA repair". Colors are randomly assigned to genes. **d** Co-IP validation of the interaction of Chd1 with selected DNA repair proteins. Figure is representative of two biologically independent experiments.

DSB labeling by terminal transferase followed by affinity purification and qPCR or deep sequencing (DSB-qPCR or DSB-seq)[17] (Fig. 3a). We first focused on rDNA (Fig. 3b) due to the mainly nucleolar accumulation of several Chd1-interacting proteins (Figs. 1 and 2), as well as the reduced rRNA synthesis and nucleolar fragmentation observed in *Chd1* KO cells[3]. Surprisingly, we found that deletion of Chd1 leads to an accumulation of DSBs at rDNA (Fig. 3c). We quantified nascent rRNA transcription by metabolic labeling of RNA with 5′-ethynyluridine (EU) coupled to biotin, followed by affinity capture and qPCR (EU-capture). Chd1 loss leads to a decrease in nascent rRNA transcripts, particularly at the 5′ end of the transcription unit (Fig. 3d). Treatment of control ES cells with RNA Pol I inhibitor CX-5461 does not induce DSB formation at rDNA (Fig. 3e), contrary to what is found in *Chd1* KO ES cells. Hence, these data indicate that DSB formation at rDNA in *Chd1* KO ES cells is not an indirect consequence of reduced Pol I transcription, further supporting the model of defective repair in the mutant cells. Moreover, the results suggest that the loss of γH2A.X in *Chd1* KO ES cells is not due to a lower level of DSBs, but rather to defective repair. A few foci of 53BP1, a marker of DNA breaks induced by exogenous damage agents, are detected at similar levels in both WT and *Chd1* KO ES cells, although they are highly increased with aphidicolin treatment (Supplementary Fig. 2c). These findings suggest that aspects of the mechanism of repair of endogenous,

transcription-induced DNA breaks are distinct from repair of DSBs induced by exogenous agents.

We previously reported that Chd1 deletion leads to global *hypo*transcription of both RNA Pol I and Pol II-transcribed genes[3]. The unexpected increase in DSBs at rDNA (Pol I-transcribed) in *Chd1* KO ES cells (Fig. 3c) led us to explore the status of DSBs at Pol II-transcribed genes, using DSB-seq (see Methods section for procedure). For comparative analyses, we also performed Chd1 and RNA Pol II ChIP-seq in *Chd1-Flag* knock-in ES cells[3].

DSB-seq confirmed the gain of breaks at rDNA (Supplementary Fig. 3a) and revealed a remarkable widespread accumulation of DSBs at promoter regions of RNA Pol II-transcribed genes in *Chd1* KO cells, relative to controls (Fig. 4a–c). DSBs occur immediately downstream of the transcription start site (TSS), where Chd1 binding peaks in wild-type (WT) conditions (Fig. 4a–c). At promoter regions (±1kb of TSS), DSBs in *Chd1* KO cells positively correlate with GC content (Spearman $\rho = 0.80$, $p < 10^{-300}$), Chd1 (Spearman $\rho = 0.71$, $p < 10^{-300}$), and RNA Pol II (Spearman $\rho = 0.63$, $p < 10^{-300}$) binding in WT cells, and negatively correlate with nucleosome occupancy (Spearman $\rho = -0.29$, $p < 5 \times 10^{-280}$). The propensity to accumulate DSBs in *Chd1* KO ES cells does not correlate with wild-type gene expression levels (Fig. 4b–d, Spearman $\rho = 0.012$, $p > 0.306$) or reduced expression upon *Chd1* loss[3] (Supplementary Fig. 3a).

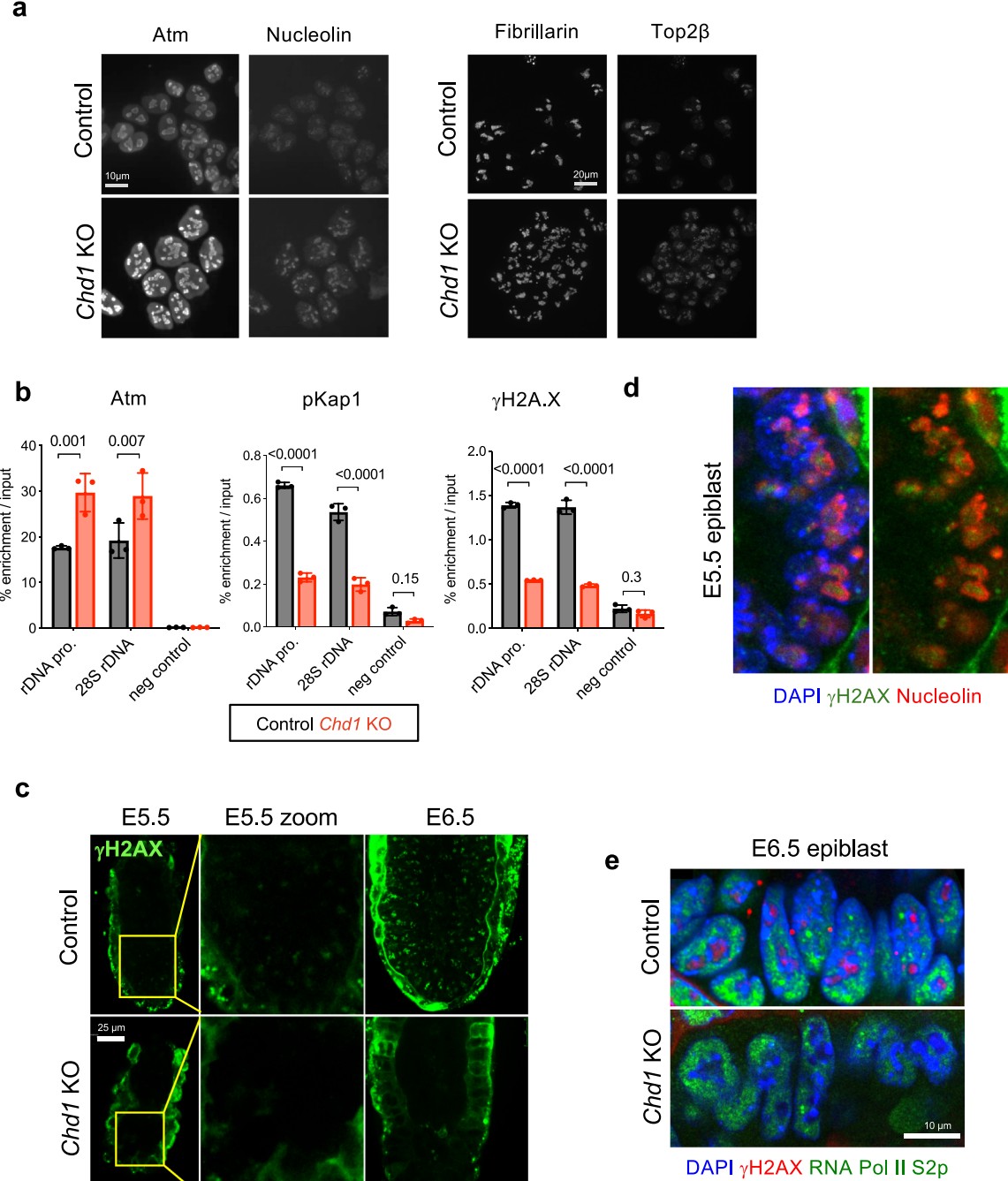

**Fig. 2 DNA repair signaling is perturbed in *Chd1* KO ES cells. a** Immunofluorescence analysis of Chd1-interacting DNA repair proteins Atm and Top2β in control and *Chd1* KO cells. Nucleolin and fibrillarin co-stainings mark the nucleoli. Note the fragmentation of nucleoli in *Chd1* KO ES cells, as we have previously reported[3]. Figure is representative of two biologically independent experiments. **b** ChIP-qPCR analysis of selected DNA repair proteins at rDNA. $N = 3$ biologically independent experiments. Graphs show mean and standard deviation. Statistical tests performed are two-way ANOVA with Sidak's correction. Values above bars indicate *p*-values. **c** γH2A.X staining of control and *Chd1* KO mouse embryos at E5.5 and E6.5. $N = 4$ (minimum) biologically independent embryos. **d** γH2A.X and Nucleolin staining of the control E5.5 epiblast, showing nucleolar localization of the γH2A.X signal. $N = 4$ (minimum) biologically independent embryos. **e** γH2A.X and elongating RNA Pol II (S2p) staining of the E6.5 epiblast in control and Chd1 KO embryos. $N = 4$ (minimum) biologically independent embryos.

We detected 5671 DSB peaks in *Chd1* KO ES cells (vs. only 54 in control), among which 1825 peaks mapped around the TSSs ($-1\text{kb}$ to $+100\,\text{bp}$) of 1785 genes (DSB-prone genes, Supplementary Data 3). DSB peaks are enriched at TSSs, 5′ untranslated regions and exons (Supplementary Fig. 3b). GO analysis predicts that genes involved in transcription, chromatin modification, and signaling are particularly prone to DSB in *Chd1* KO ES cells

(Supplementary Fig. 3c). To understand why these genes might be especially susceptible to DNA breaks, we analyzed the chromatin structure at DSB-prone TSSs in comparison to non-DSB-prone TSSs. We utilized ES cell MNase-seq datasets generated by Voong et al.[19] (Fig. 4d). DSB-prone genes display a more open chromatin structure at TSSs, with less regular nucleosomes surrounding the start site. In particular, the +1 nucleosome is less

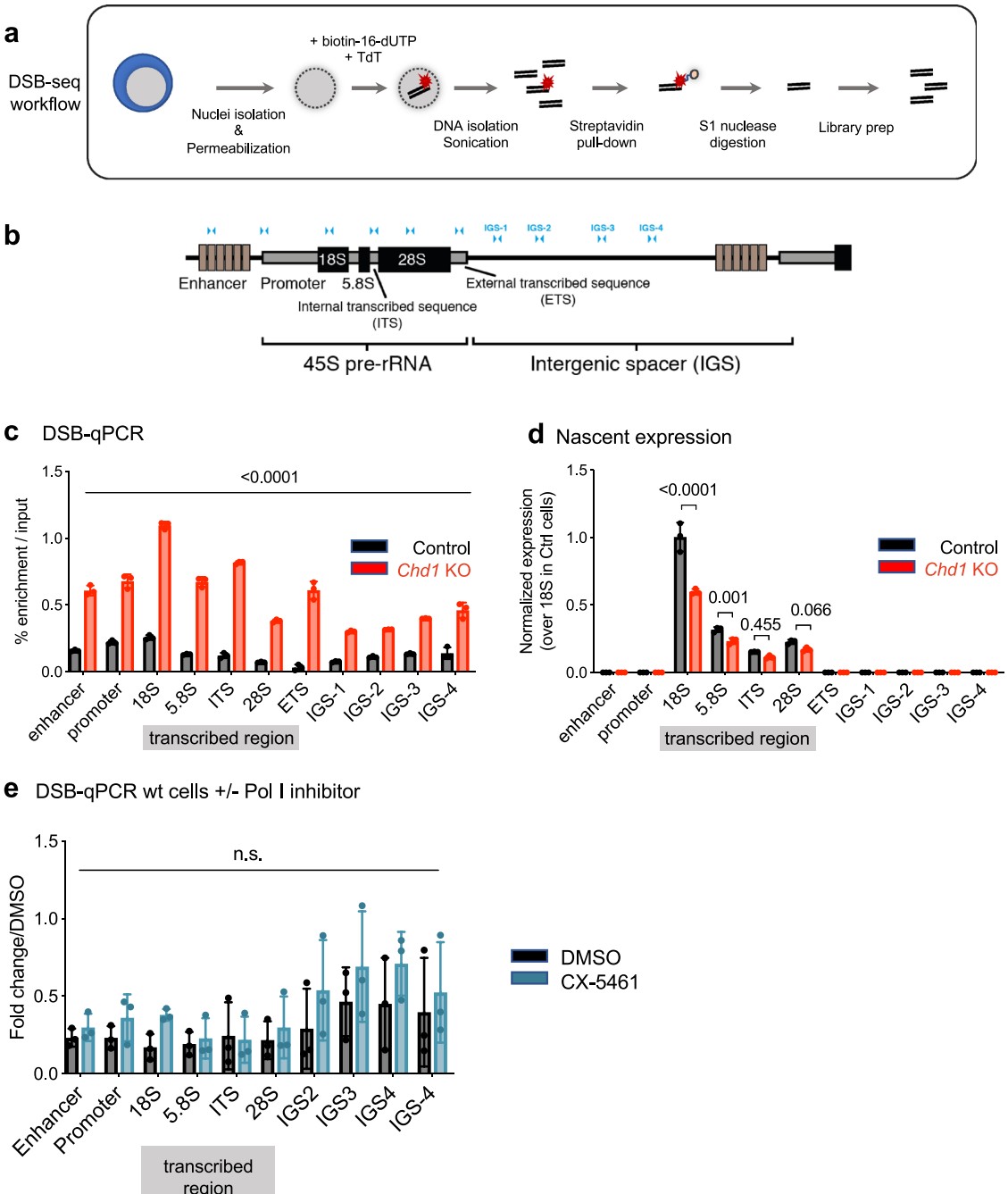

**Fig. 3 Double-stranded breaks accumulate at ribosomal DNA in *Chd1* KO cells. a** Schematic depiction of the workflow of DSB-PCR/seq. TdT terminal deoxynucleotidyl transferase, DSB double-stranded break. **b** Schematic depiction of one rDNA transcription unit. **c** DSB levels along the rDNA unit in control and *Chd1* KO cells, normalized to input. $N = 3$ biologically independent experiments. Graphs show mean and standard deviation. Statistical tests performed are two-way ANOVA with Sidak's correction. Values above bars indicate *p*-values. **d** Nascent rRNA transcription in control and *Chd1* KO cells. $N = 3$ biologically independent experiments. Graphs show mean and standard deviation. Statistical tests performed are two-way ANOVA with Sidak's correction. Values above bars indicate *p*-values. **e** DSB levels in wt ES cells with or without treatment with a Pol I inhibitor. $N = 3$ biologically independent experiments. Graphs show mean and standard deviation. Statistical tests performed are two-way ANOVA with Sidak's correction. Values above bars indicate *p*-values.

abundant in DSB-prone genes (Fig. 4d). Taken into consideration that this graph depicts a population average of many cells going through the transcription cycle, the data suggest that the +1 nucleosome in DSB-prone genes is frequently displaced to expose naked DNA. Interestingly, Chd1 peaks at this +1 position (Fig. 4a), suggesting that it promotes eviction of this nucleosome, as previously reported in embryonic fibroblasts[10], and in doing so facilitates DNA repair and transcriptional elongation.

To probe why these 1785 genes have a more exposed TSS region and are DSB-prone upon Chd1 loss, we investigated genomic features of these genes. DSB-prone genes have a significantly higher GC content in their promoter-proximal region compared to non-DSB-prone genes (Fig. 4a, e, Wilcoxon rank-sum test $p < 10^{-300}$). Moreover, DSB-prone genes are on average longer than non-DSB-prone genes (Fig. 4f, Wilcoxon rank-sum test $p < 1.62 \times 10^{-99}$). Previous studies showed that longer genes

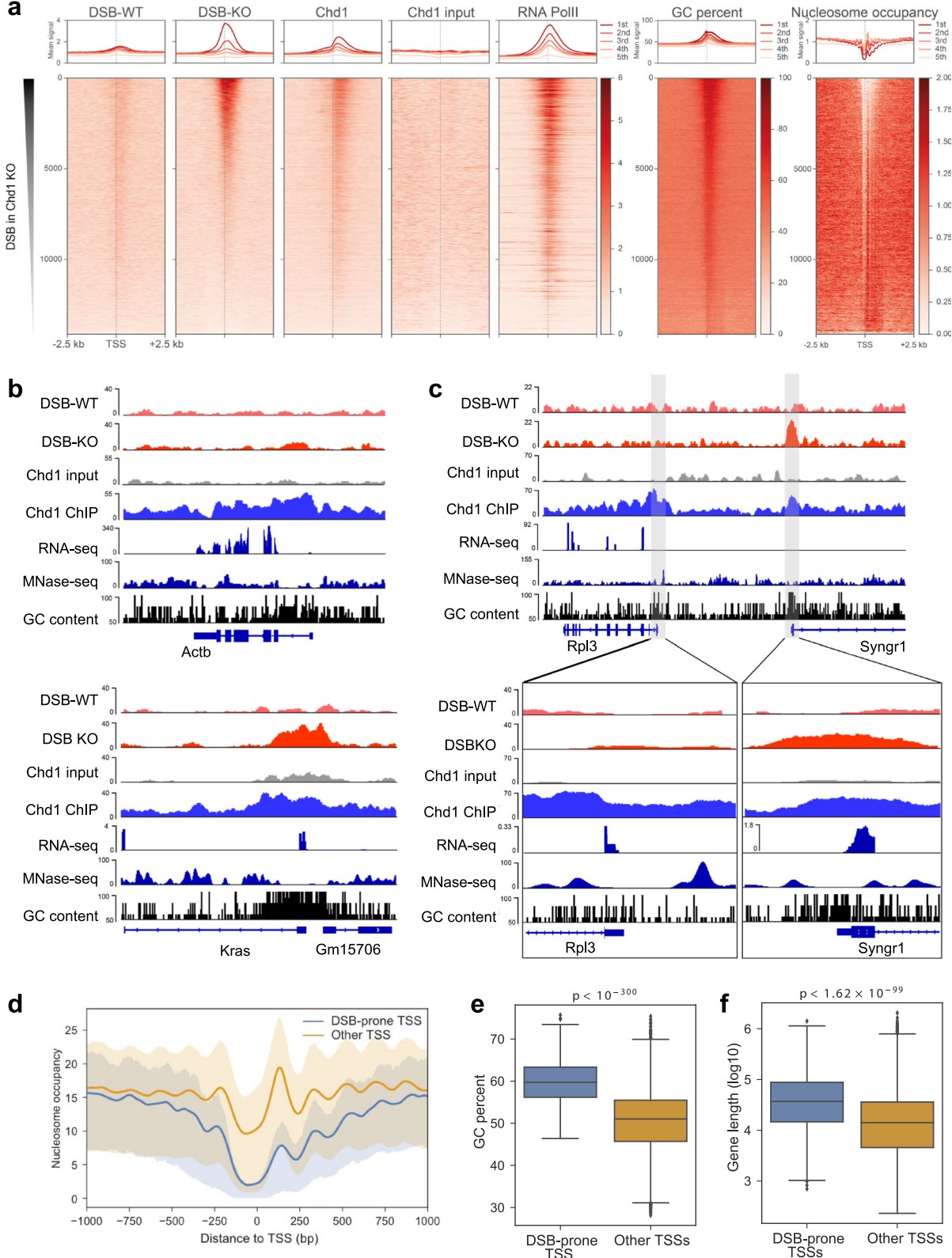

are more DSB-prone in neurons, due to the increased Topoisomerase activity required to relieve the torsional stress that accumulates during DNA unwinding (reviewed in Teves et al.[16]). Taken together, our results suggest that Chd1 remodels nucleosomes at GC-rich promoters of long genes to facilitate DNA repair and repeated cycles of RNA Pol II elongation in hypertranscribing ES cells.

Finally, we explored the relationship between DSBs induced by Chd1 loss in ES cells and the activities of Topoisomerase, transcription initiation, and DNA repair. Topoisomerase creates then ligates DSBs, unless the cells are treated with etoposide which blocks its ligation activity. Treatment of control cells with etoposide increases DSB formation, indicating that the DSBs do occur in wt cells but are promptly repaired by Topoisomerases in

**Fig. 4 Widespread double-stranded break accumulation at transcriptional start sites (TSSs) in *Chd1* KO cells. a** Heatmaps showing DSBs, Chd1 binding, Pol II levels, nucleosome occupancy, and GC content. Genes are ranked based on mean double-stranded break (DSB) levels in *Chd1* KO cells within ±1kb of TSS. Values in each heatmap are divided by the mean of the entire matrix (except for GC content), respectively, and then smoothed using a Gaussian kernel (5 genes by 20 bp), to facilitate visualization. Upper panels show the mean signal within each quintile of genes sorted according to the same order as in the heatmaps. Only protein-coding genes with unique TSSs are included to avoid ambiguity. MNase-seq data were obtained from Voong et al.[19]. **b, c** Genome browser views of DSB-prone (Kras, Syngr1) and other (Actb, Rpl3) genes. Note that DSB propensity does not correlate with expression level (see Supplementary Fig. 3a). **d** Nucleosomal patterns at TSSs of DSB-prone and non-DSB-prone genes. Line depicts mean and shade shows first-to-third quartile values. **e** GC content of DSB-prone TSSs in comparison to non-DSB-prone TSSs. Statistical test is one-sided Wilcoxon rank-sum test. **f** Gene length of DSB-prone genes in comparison to non-DSB-prone genes. Statistical test is one-sided Wilcoxon rank-sum test. In **d–f**, only protein-coding genes with unique TSSs are included.

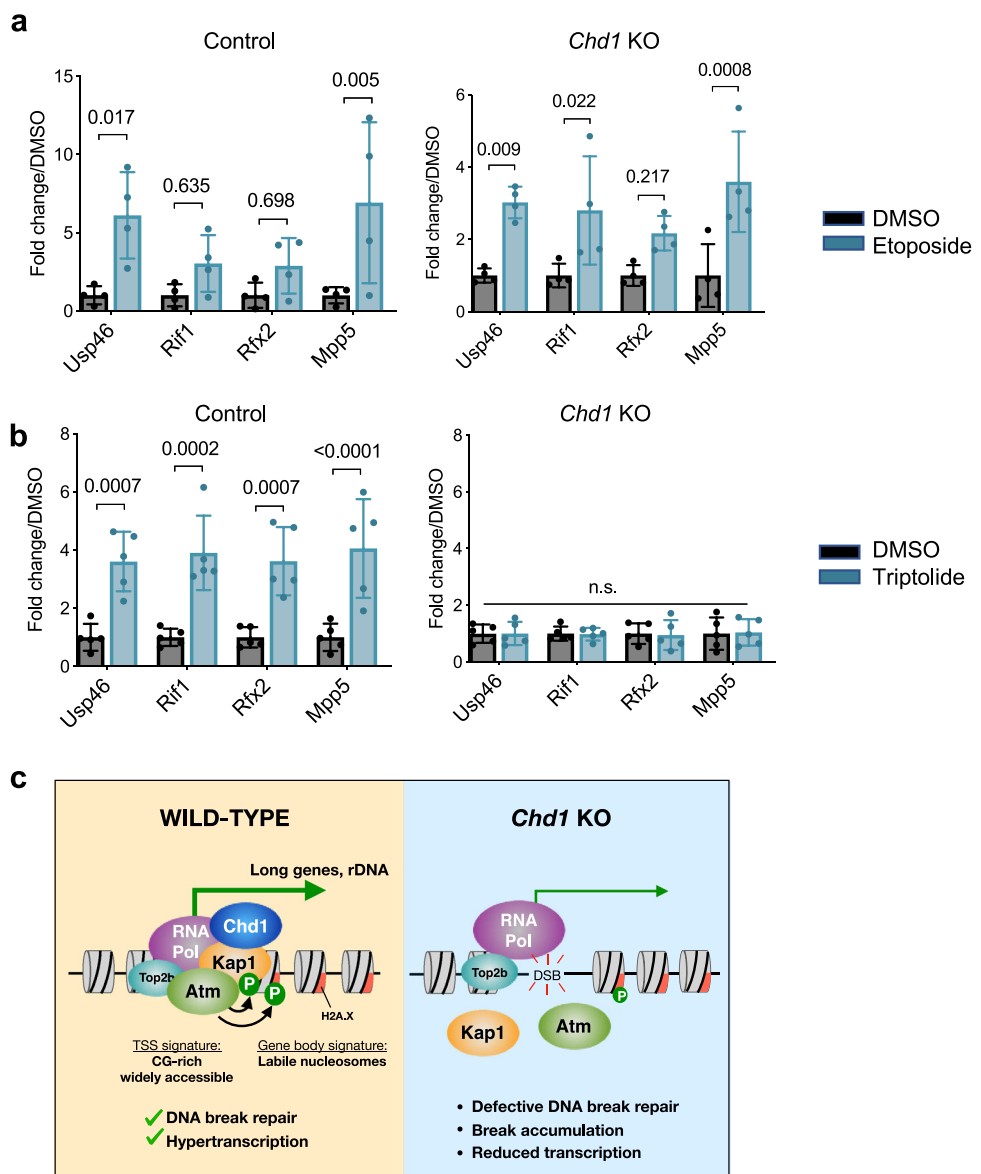

**Fig. 5 Promoter-proximal DSBs occur independent of transcription initiation but are dependent on topoisomerase activity. a** Gene promoter DSB levels in control and *Chd1* KO cells with or without treatment of etoposide. $N = 4$ biologically independent experiments. Graphs show mean and standard deviation. Statistical tests performed are two-way ANOVA with Sidak's correction. Values above bars indicate *p*-values. **b** Gene promoter DSB levels in control and *Chd1* KO cells with or without treatment of triptolide. $N = 4$ biologically independent experiments. Graphs show mean and standard deviation. Statistical tests performed are two-way ANOVA with Sidak's correction. Values above bars indicate *p*-values. **c** Proposed model for how Chd1 may protect genome integrity at transcribed promoters by preventing DNA break accumulation during hypertranscription in ES cells. See text for details.

the presence of Chd1 (Fig. 5a and Supplementary Fig. 4a). Etoposide treatment further increases DSB levels in *Chd1* KO cells, suggesting that the DSBs observed in *Chd1* KO cells are not simply due to defective Topoisomerase activity, a finding that

requires further investigation. Overall, inhibition of topoisomerase activity can induce DSBs at active promoters, as has been shown previously[17], and this is the case even in the absence of Chd1.

Although DSB occurrence in *Chd1* KO cells does not correlate with gene expression levels in wt cells or with differential expression in *Chd1* KO cells, DSBs do correlate to some extent with RNA Pol II occupancy at promoters (Fig. 4a). We note that this correlation is not linear, as the 1st and 2nd quartiles of highest DSB-accumulating genes have a greater difference in DSB levels than Pol II occupancy in control cells. Considering the fact that Chd1 associates with Pol II and facilitates transcription just downstream of Pol II through pumping DNA towards it[20], *Chd1* KO could compromise transcription via accumulation of initiated Pol II at TSSs. To probe the role of transcription initiation on DSB occurrence, we tested DSB levels upon inhibition of transcription initiation via triptolide treatment (Fig. 5b and Supplementary Fig. 4b). Inhibition of transcription initiation does increase DSBs in wt cells to levels comparable to *Chd1* KO cells but does not further increase DSBs in *Chd1* KO cells. These data suggest that DSBs that form in *Chd1* KO ES cells may occur during Pol II loading and assembly, while reduced nascent transcription may further contribute to DSBs accumulation.

## Discussion

This work describes a novel role for the chromatin remodeler Chd1 in protecting genome integrity at promoter regions by preventing DSB accumulation in pluripotent stem cells (Fig. 5c). Our results point to a central role for rDNA at the nucleolus in the coordination between hypertranscription and DNA integrity in hypertranscribing ES cells. rRNA comprises ~80% of the RNA being synthesized in ES cells and therefore represents both a major focal point of hypertranscription as well as a vulnerability to DNA breaks. We had previously implicated Kap1, generally thought to be a repressor, in rRNA transcription[21], and this study further contributes to clarifying that role.

We propose that the accumulation of unrepaired DNA breaks in *Chd1* KO cells compromises nascent transcriptional output and leads to the proliferation defects and ultimate developmental arrest of the rapidly expanding post-implantation epiblast. We note that the loss of Chd1 is not catastrophic to ES cells, despite the widespread occurrence of DSBs at GC-rich promoters of transcribed genes: *Chd1* KO ES cells remain undifferentiated and capable of gene transcription, albeit with a lower transcriptional output and a self-renewal deficit. These results suggest that promoter DSBs are still eventually repaired in *Chd1* KO ES cells, albeit at a lower rate, which compromises optimal nascent transcription and proliferation. Although we did not observe a direct correlation between DSB formation and expression levels in wt or expression changes in *Chd1* KO cells, a role for reduced transcriptional output on DSB formation via RNA Pol II accumulation or pausing is a possibility that warrants further studies. Similarly, understanding to what extent Chd1 specifically suppresses DSB formation in the first place versus promotes their rapid repair will require more detailed biochemical studies. These functions may not be mutually exclusive. For example, it is possible that Chd1 facilitates the religation activity of Top2a, and in parallel recruits DNA repair factors as a backup mechanism that may be particularly important in hypertranscribing cells. In either scenario, Chd1 activity at transcribed promoters protects genome integrity, a finding with significant implications in stem cell biology and cancer.

Repair of DSBs requires efficient signaling spearheaded by the Atm kinase, which we show is defective in *Chd1* KO ES cells. Interestingly, Chd1-deficient human cancer cells have defective DNA repair when exposed to ionizing radiation or radiomimetic chemicals[22,23]. These cells show reduced H2A.X phosphorylation and are more sensitive to PARP inhibitors. The present study is the first report of Chd1-mediated DNA repair activity under native conditions without any external insult. The TSS sequence and chromatin landscape appear to be the main determinants of DSB generation in ES cells. DSBs were shown to be conducive to transcription in several cell types, in the absence of exogenous DNA damage[11–13,17]. Such studies point to a correlation between gene transcription and the occurrence of DNA breaks. *Chd1* KO ES cells, on the other hand, have lower nascent transcription levels and higher incidence of DNA breaks. Thus, the DNA breaks we observe in *Chd1* KO ES cells are not caused by increased transcription but rather by defective repair, a notion further supported by the loss of γH2A.X. We speculate that hypertranscribing, proliferating cells may have an increased dependence on Chd1 to balance high transcriptional output with DNA integrity. It will be of interest to explore the potential role of Chd1 in transcription-associated DNA break repair in other cell types, both in physiological stem/progenitor cells as well as in proliferating tumor cells. Of note, CHD1 is the second most frequently mutated gene in prostate cancer, after PTEN[24,25]. In agreement with our data, Dellino et al.[26] showed that release of paused RNA Pol II in breast cancer cells induces DSBs preferentially at long genes and can lead to chromosomal translocations.

In highly proliferative cells, replication stress can occur due to the collision of replication and transcription complexes, resulting in DNA breaks that are marked by γH2A.X[27]. It is unlikely that the high γH2A.X signal in wild-type ES cells and epiblast is strictly due to replicative stress because high γH2A.X is observed in all cells regardless of cell cycle stage. Moreover, *Chd1* KO ES cells have higher incidence of DSBs and reduced γH2A.X (this study) without significant deviations in cell cycle stage proportions[3]. In agreement, γH2A.X staining is not limited to S phase cells, as might be expected from replication stress, but is found across all stages of the cell cycle, in both control and *Chd1* KO ES cells (Supplementary Fig. 4c–e). Taken together, our findings suggest that hypertranscription puts ES cells, and potentially other stem/progenitor cells, at risk of DNA breaks and genomic instability, but this is countered by Chd1-dependent DNA repair.

Why Atm signaling is defective in *Chd1* KO ES cells is a central question that remains elusive. In *Chd1* KO cancer cells, γH2A.X is reduced due to reduced incorporation or retention of H2A.X at damage sites[23]. Decreased γH2A.X despite DSB accumulation could stem from defective γH2A.X spreading due to compromised H2A.X deposition in *Chd1* KO ES cells. H2A.X is deposited by the FACT complex[28], which interacts with Chd1 to spread through transcribed regions[29]. Therefore it is conceivable that the amplification of the DNA damage signal through γH2A.X spreading is defective in *Chd1* KO cells. Alternatively, it is possible that the chromatin remodeling activity of Chd1 facilitates access of ATM to its target site on H2A.X, or that other interactors of Chd1 promote Atm activity towards H2A.X. For example, Chd1 may also interact with PARP1 and the histone acetyltransferase Tip60, and both histone ADP ribosylation and acetylation are involved in chromatin relaxation at DNA repair sites[30,31]. Other major transcriptional regulators such as Myc and Paf1c mediate repair of transcription-associated DNA breaks through H2B ubiquitination[32], which may be a mechanism shared by Chd1, as global H2Bub reduction was observed in *Chd1* KO cells[33]. Further biochemical studies of how the activity of Atm and other aspects of DNA repair are modulated by Chd1 will shed light on the mechanisms by which stem and progenitor cells can undergo hypertranscription while preserving DNA integrity.

## Methods

**Mice**. Chd1$^{\Delta/+}$ females (6- to 12-week-old) and males (6 week- to 6-month-old) were used to recover the embryos. Animals were maintained on 12 h light/dark

cycle and provided with food and water *ad libitum* in individually ventilated units (Techniplast at TCP, Lab Products at UCSF) in the specific pathogen-free facilities at UCSF. All procedures involving animals were performed in compliance with the protocol approved by the IACUC at UCSF, as part of an AAALAC-accredited care and use program (protocol AN091331-03). Chd1 heterozygous mice were mated, embryos were collected at embryonic day 5 (E5.5) or day 6 (E6.5) after detection of the copulatory plug by dissecting uteri of pregnant females following standard practices.

**ES cell culture**. *Chd1-Flag* knock-in, *Chd1*$^{fl/\Delta}$ and *Chd1*$^{\Delta/\Delta}$ ES cells were used as described[3]. Cells were grown in DMEM GlutaMAX with 15% FBS (Atlanta Biologicals), 0.1 mM non-essential amino acids, 50 U/ml Penicillin/Streptomycin (UCSF Cell Culture Facility), 0.1 mM EmbryoMax 2-Mercaptoethanol (Millipore), and 2000 U/ml ESGRO supplement (LIF, Millipore or Gemini) under ambient air with 5% $CO_2$. Cells tested negative for mycoplasma contamination.

**Immunoprecipitation**. Wild-type E14 or *Chd1-Flag* knock-in cells were used. Cells were fractionated into cytoplasmic and nuclear compartments prior to immuno-precipitation. For cytoplasmic extracts, cells were lysed in Buffer A (10 mM Hepes pH 7.9, 5 mM $MgCl_2$, 0.25 M Sucrose and 0.1% NP-040 supplemented with 1x Halt protease inhibitor cocktail (Thermo Fisher Scientific, 78425), 1 mM PMSF, 5 mM NaF and 1 mM NaVO$_4$) and were centrifuged for 10 min, 1800×*g* at 4 °C. For nuclear extracts, the resulting pellets were resuspended in buffer B (10 mM Hepes pH 7.9, 1 mM MgCl2, 0.1 mM EDTA, 25% glycerol, 0.5% Triton X-100 and 0.5 M NaCl supplemented with 1x Halt protease inhibitor cocktail (Thermo Fisher Scientific, 78425), 1 mM PMSF, 5 mM NaF and 1 mM NaVO$_4$) and homogenized by passing through 18Gx1 1/2" size needles. Nuclear extracts were quantified using Pierce BCA Protein Assay kit (23225). For immunoprecipitation, 100 μg of extract was adjusted to 500 μl total volume and 150 mM final NaCl concentration, and incubated in the presence of 20 μl pre-washed Protein A or G Dynabeads (Thermo Fisher Scientific, 1002D or 1004D) and the following antibodies: Flag (Sigma, F1804), Kap1 (Abcam, ab22553), Atm phospho S1981 (Active Motif, 39529), Atr (Santa Cruz Biotechnology, sc-1887), Parp1 (Santa Cruz Biotechnology, sc-25780), Top2β (Santa Cruz Biotechnology, sc-13059), and Xrcc1 (Santa Cruz Biotechnology, sc-11429). Beads were washed three times in buffer B, then boiled in 2x Laemmli Buffer with 5% β-mercaptoethanol. Western blot was performed as described below.

**IP- mass spectrometry**. Chd1-Flag ES cells were lysed and processed as above. Nuclear extracts were used for immunoprecipitation at 150 mM and 250 mM salt concentrations. For mass spectrometry, Flag IP and IgG control were run on a denaturing gel. Gel was stained using Coomassie Brilliant Blue. IP and control lanes were cut into 10 pieces avoiding light and heavy antibody chains and frozen until processed for mass spectrometry.

Protein bands were excised, destained with repeated incubation in 200 mM ammonium bicarbonate, 40% [v/v] acetonitrile. Gel pieces were dried with three washes in 100% acetonitrile and then trypsinised (Trypsin resuspended in 100 mM ammonium bicarbonate, 5% [v/v] acetonitrile) overnight at 37 °C. Peptides were extracted from the gel pieces by incubation in 50% [v/v] acetonitrile, 0.1% [v/v] formic acid, peptides were desiccated and resuspended in 2% [v/v] acetonitrile, 0.05% [v/v] trifluoroacetic acid; pH 2.7. For each analysis, 10% of the peptide sample was loaded onto an Acclaim Pepmap C18 Trap (500 μm × 5 mm) and flow was set to 30 μl/min of 2% [v/v] acetonitrile, 0.05% [v/v] trifluoroacetic acid for 5 min. Analytical separation of the peptides was performed using Acclaim PepMap100C18 Column (3 μm, 75 μm × 500 mm) on a U3000 RSLC (Thermo). Briefly, peptides were separated over a 91 min solvent gradient from 2% [v/v] acetonitrile, 0.1% [v/v] formic acid to 40% [v/v] acetonitrile, and 0.1% [v/v] formic acid on-line to a LTQ Orbitrap Velos (Thermo). Data was acquired using an data dependant acquisiton (DDA) method where, for each cycle one full MS scan of *m/z* 300–1700 was acquired in the Orbitrap at a resolution of 60,000 at *m/z* 400 with an AGC target of 1x10e6. Each full scan was followed by the selection of the 20 most intense ions, CID and MS/MS analysis was performed in the LTQ. Selected ions were excluded from further analysis for 60 s. Ions with an unassigned charge or a charge of +1 were rejected.

Data were analyzed using Mascot (Matrix Sciences) the parameters were; Uniprot database, taxonomy *Mus Musculus*, 16,403 sequences present, trypsin with up to 1 missed cleavage allowed, variable modification were oxidized methionine, phosphorylated serine, threonine, and tyrosine and the peptide tolerance of 0.025 Da and 0.03 Da for MS/MS tolerance. The false discovery rate (FDR) of protein identification was controlled using a target-decoy searching strategy[34]. The maximum allowed FDR for protein identification was set to 1%. The posterior error probability (PEP) score denotes the probability that the identified peptide is correct.

For gene ontology (GO) and protein network analysis, proteins with fewer than 4 peptides in Chd1 IP were filtered out and only those hits passing the log2FC (Chd1 IP/negative control IP) >2 and PEP score <0.05 were retained. Gene ontology (GO) pathway enrichment analysis was performed using the compareCluster function of clusterProfiler[35] (version 3.18.1). The "enrichGO" function for biological process with a significance cutoff *p*-value < 0.05 and *q*-value

< 0.1 was used. Benjamini–Hochberg method was used to correct for multiple statistical testing (adjusted *p*-values).

Protein network analysis was performed using the STRING software[36] (version 11.0) with the following parameters: Network type: Full STRING network, meaning of network edges: confidence (line thickness indicates the strength of data support), active interaction sources: experiments, textmining, databases, co-expression, neighborhood, gene fusion, co-occurrence, minimum required interaction score = 0.4.

**Western blot analysis**. *Chd1*$^{fl/\Delta}$ and *Chd1*$^{\Delta/\Delta}$ (7 days post-induction of KO) ES cells were used. For analysis of whole cell extracts, cells were lysed in RIPA buffer (150 mM NaCl, 1% NP-40, 0.5% sodium deoxycholate, 0.1% SDS, 50 mM Tris pH 8.0, 1x Halt protease inhibitor cocktail (Thermo Fisher Scientific), 1 mM PMSF, 5 mM NaF and 1 mM NaVO$_4$). Cells were incubated for 30 min on ice, then sonicated using a Bioruptor (Diagenode) for 5 min with settings high, 30 s on, 30 s off. Laemmli Buffer with 5% β-mercaptoethanol was added to 1x and samples were boiled at 95 °C for 5 min. Extracts were loaded into 4–15% Mini-Protean TGX SDS Page gels (Bio-Rad). Proteins were transferred to PVDF membranes. Membranes were blocked in 5% milk/PBS-T buffer for 30 min and incubated either overnight at 4 °C or for 1 h at room temperature with the following antibodies: Chd1 (1:1000, Cell Signaling, 4351), Top2β (1:500, Santa Cruz Biotechnology, sc-13059), Atm (1:250, Genetex, GTX70103 and Abcam, ab78), p-Atm (1:250, Thermo, MA1-2020), Kap1 (Abcam, ab22553), Nucleolin (1:1000, Abcam, ab22758), Polr1a (1:1000, Cell Signaling, D6S6S), H2A.X (1:2500, Abcam, ab11175), γH2A.X (1:1000, Abcam, ab2893), Parp1 (1:500, Santa Cruz Biotechnology, sc-25780), Gapdh (1:2000, Millipore, MAB-374), and anti-rabbit/mouse/goat secondary antibodies (1:2000, Jackson Labs, 115-035-062, 111-035-144). Membranes were incubated with ECL or ECL Plus reagents and exposed to X-ray films (Thermo Fisher Scientific).

**Immunofluorescent staining and imaging**. Chd1$^{fl/\Delta}$ and Chd1$^{\Delta/\Delta}$ ES cells were used. Cells were plated on matrigel in 8-chamber polystyrene vessels. Cells were fixed in 4% paraformaldehyde for 10 min, washed with PBS, and permeabilized with 0.2% Triton X-100 in PBS for 5 min on ice. After blocking in PBS, 2.5% BSA, 5% donkey serum for 1 h, cells were incubated overnight at 4 °C with the following antibodies: γH2A.X (1:500, Abcam, ab2893), Nucleolin (1:1000, Abcam, ab22758), Atm (Genetex, GTX70103), Top2β (1:200, Santa Cruz Biotechnology, sc-13059), 53BP1 (1:500, Abcam, ab175933), and Fibrillarin (1:200, Abcam, ab4566). Cells were washed in PBS-Tween20, 2.5% BSA, incubated with fluorescence-conjugated secondary antibody (Life Technologies) for 2 h at room temperature and mounted in VectaShield mounting medium with DAPI (Vector Laboratories). Imaging was performed using a Leica BL-23 microscope. Staining and imaging of E5.5 and E6.5 embryos were performed as described in Guzman-Ayala et al.[3]. Briefly, embryos were fixed in 4% paraformaldehyde in PBS overnight at 4 °C. For IF, embryos or cells were permeabilized with 0.5% Triton X-100 for 20 min at room temperature and blocked in 0.5% BSA in PBS. Embryos were imaged on a laser-scanning inverted confocal microscope (CTR 6500, Leica). z-stacks were taken at 5 μm intervals through the embryo, with each channel acquired sequentially. A minimum of 3 embryos was used for each experiment.

**Chromatin immunoprecipitation**. Chd1$^{fl/\Delta}$ and Chd1$^{\Delta/\Delta}$ (7 days post-induction of KO) ES cells were used. ChIP was performed as described in Brookes et al.[37] with modifications: after aspiration of culture medium, cells were washed with PBS and fixed on the culture dish using 1% formaldehyde in PBS for 10 min at room temperature (RT). Glycine was added to a final concentration of 125 mM to quench formaldehyde for 5 min at RT. Cells were washed twice with ice-cold PBS, incubated in Swelling Buffer (25 mM HEPES pH 7.9, 1.5 mM MgCl$_2$, 10 mM KCl, 0.1% NP-40 with 1x Halt protease inhibitor cocktail (Thermo Fisher Scientific, 78425), 1 mM PMSF, 5 mM NaF and 1 mM NaVO$_4$) for 10 min, scraped, passed through an 18Gx11/2″ needle (5x) and spun down at 3000×*g*, 4 °C, 5 min. Nuclei were resuspended in Sonication Buffer (50 mM HEPES pH 7.9, 140 mM NaCl, 1 mM EDTA, 1% Triton X-100, 0.1% Na-deoxycholate 0.1% SDS with 1x Halt protease inhibitor cocktail, 1 mM PMSF, 5 mM NaF, and 1 mM NaVO$_4$) and sonicated using a Covaris S2 sonicator with settings 5% duty cycle, intensity 4, cycles per burst 200, frequency sweeping. In all, 20 μl chromatin was incubated sequentially with 1 μl RNaseA and 5 μl proteinase K in 100 μl total volume at 37 °C for 30 min and 65 °C for 1 h, purified using a Qiagen PCR purification kit and DNA content was quantified using a NanoDrop. Fragment size distribution was checked on a 1% agarose gel. Chromatin was snap frozen if not immediately used for IP. Chromatin volume equivalent to 25 μg DNA was used for each IP. Chromatin was immunoprecipitated in the presence of 20 μl pre-washed Protein A or G Dynabeads (Thermo Fisher Scientific, 1002D or 1004D) and the following antibodies: Flag (Sigma, F1804), Kap1 (Abcam, ab22553), phospho S824 Kap1 (Abcam, ab70369), γH2A.X (Abcam, ab2893), H2A.X (Abcam, ab11175), H1 (Thermo Fisher Scientific, PA128374), Polr1a (Cell Signaling, D6S6S), and Atm (Genetex, GTX10701). Beads were washed in sonication buffer (2 times), wash buffer A (sonication buffer with 500 mM NaCl) and TE buffer (10 mM Tris pH 8.0, 1 mM EDTA), and resuspended in elution buffer (50 mM Tris pH 7.5, 1 mM EDTA, 1% SDS) with 1 μl RNaseA and 5 μl proteinase K in 100 μl total volume. After incubation at 37 °C for

30 min and 65 °C for 2 h to overnight, DNA was purified using a Qiagen PCR purification kit. qPCR was performed with KAPA SYBR FAST qPCR Master Mix (Kapa Biosystems) and amplified on a 7900HT Real-time PCR machine (Applied Biosystems). For sequencing, libraries were prepared using the NEBNext ChIP-seq Library Prep for Illumina kit. Library quality and quantity were analyzed using Bioanalyzer (Agilent). Samples were sequenced on a HiSeq 2500 using single-end 50 bp sequencing reads in rapid mode.

**Detection of double-stranded DNA breaks.** Chd1$^{fl/\Delta}$ and Chd1$^{\Delta/\Delta}$ (established) ES cells were used. For detection of double-stranded DNA breaks, we combined the Baranello et al.[17] and BLESS protocols[38] to achieve end labeling in situ. $1.5 \times 10^8$ cells were fixed in 2% formaldehyde suspension for 30 min at room temperature, followed by quenching in 0.25 M glycine for 5 min. Cells were centrifuged, washed twice in PBS, and incubated in 25 ml lysis buffer (10 mM Tris pH 8.0, 10 mM NaCl, 1 mM EDTA, 1 mM EGTA, 0.2% NP-40) for 1 h at 4 °C. Pellets were resuspended in nucleus break buffer ((10 mM Tris pH 8.0, 150 mM NaCl, 1 mM EDTA, 1 mM EGTA, 0.3% SDS, 1 mM DTT) and incubated for 45 min at 37 °C. After centrifugation, nuclei were resuspended in 1x NEB buffer 2 on ice. In all, 10 μl Proteinase K (20 mg/ml) was added, digestion was performed for 4 min at 37 °C, and samples were immediately returned to ice and 25 μl PMSF was added. Nuclei were centrifuged, and resuspended in 5 ml 1x NEB Buffer 2 + 15 μl Triton X-100, and centrifuged again at 200×g for 10 min at 4 °C. Nuclei were washed once in water, and divided into two tubes for end labeling and control reactions. Nuclei were resuspended in 625 μl 1x TdT buffer with 2.5 μl TdT (Promega, M1871) and 2 μl biotin-16-dUTP (Roche 11093070910), and were incubated at 37 °C for 1 h. Control reaction was assembled the same way without TdT. 1:50 volume of 0.5 M EDTA was added to stop the reaction. Proteins were digested with 5 μl Proteinase K (20 mg/ml) at 2 h to overnight at 65 °C. Labeled DNA was precipitated using sodium acetate (70 μl) and isopropanol (700 μl), centrifuged, washed with 70% ethanol and resuspended in water. Genomic DNA was sonicated using a Covaris S2 sonicator with settings 5% duty cycle, intensity 4, cycles per burst 200, frequency sweeping for 4.5 cycles. DNA amount was quantified on a NanoDrop. Size distribution was checked on a 1% agarose gel. For biotinylated DNA pull-down, 20 μg DNA was adjusted to 600 μl final volume in W&B buffer (10 mM Tris pH 7.5, 1 mM EDTA, 1 M NaCl). In all, 10 μl Dynabeads MyOne C1 was washed twice with W&B buffer and added to DNA. Sample was incubated for 1 h at 4 °C. Beads were washed twice in W&B buffer, and resuspended in 100 μl elution buffer (95% v/v formamide, 10 mM EDTA). After incubation at 65 °C for 5 min, DNA was purified using a Qiagen PCR purification column. For sequencing, biotinylated overhangs were removed using S1 nuclease. Sequencing libraries were prepared using the NEBNext ChIP-seq Library Prep for Illumina kit. Library quality and quantity were analyzed using Bioanalyzer (Agilent). Samples were sequenced on a HiSeq 4000 using single-end 50 bp sequencing reads. A list of primers used for qRT-PCR analysis is available in Supplementary Table 1.

**Nascent RNA capture followed by qRT-PCR.** To measure nascent transcriptional changes at specific loci, ES cells were analyzed using the Click-iT Nascent RNA Capture Kit (Life Technologies). Chd1$^{fl/\Delta}$ and Chd1$^{\Delta/\Delta}$ cells were incubated with 0.2 μM 5-ethynyl uridine (EU) for 30 min to label nascent transcripts. Cells were washed, harvested by trypsinization and counted. Total RNA was isolated from the $10^6$ Chd1$^{fl/\Delta}$ or Chd1$^{\Delta/\Delta}$ cells using the Qiagen RNeasy Mini Kit (Qiagen) and processed according to manufacturer's instructions. qPCR was performed with KAPA SYBR FAST qPCR Master Mix (Kapa Biosystems) and amplified on a 7900HT Real-time PCR machine (Applied Biosystems).

**Inhibitor treatments.** Chd1$^{fl/\Delta}$ and Chd1$^{\Delta/\Delta}$ (established) ES cells were used. Cells were incubated for indicated durations and concentrations with Triptolide (Sigma), Etoposide (Sigma), or CX-5461 (Sigma). Control cells were treated with DMSO. Inhibitors were withdrawn and cells were washed immediately prior to detection of double-stranded DNA breaks and qPCR.

**Cell cycle analysis.** *Chd1$^{fl/\Delta}$* and *Chd1$^{\Delta/\Delta}$* (established) ES cells were used. Overnight cultured cells were incubated with 10 μM 5-ethynyl-2′-deoxyuridine (EdU) for 1 h. Cells were harvested, fixed in 4% PFA for 15 min, and permeabilized in 0.25% Triton-X 100 in PBS for 5 min on ice. Cells were blocked with 1% BSA in PBS and incubated in γH2A.X primary antibody (Abcam, ab2893) for 20 min at room temperature. Following a second wash, cells were incubated with fluorescence-conjugated secondary antibody (Life Technologies) for 20 min at room temperature. Subsequent EdU labeling was conducted according to manufacturer instructions of the Click-iT EdU Alexa Fluor 488 Flow Cytometry Assay Kit ((Life Technologies). SYTOX Blue (Invitrogen) was used to detect DNA content at 1:1000 dilution. Data was collected on a Sony MA900 Multi-Application Cell Sorter, analyzed using FCS Express 7, and plotted using Prism 7.

**Bioinformatic analyses**

*ChIP-seq, DSB-seq, and MNase-seq data processing.* ChIP-seq and DSB-seq reads were mapped to the mm9 genome using Bowtie2[39] with the options --end-to-end --sensitive --score-min L,-1.5,-0.3. Reads with mapping quality <13 were filtered out. PCR duplicates were removed by keeping at most one mapped read at each

genomic position. Biological replicates were then combined. Read coverage profiles were generated using bedtools[40] after extending the mapped reads from 5′ to 3′ end to 200 bp. Processed MNase-seq data in mouse ES cells were obtained from Voong et al[19]. with accession number GSE82127. Center-weighted nucleosome occupancy was calculated from the provided nucleosome center scores as described in Voong et al. and used throughout the paper.

*DSB-seq peak calling and annotation.* DSB-seq peaks were called using MACS2[41] with the options --shift 0 --nomodel --extsize 200 -g mm and default q-value cutoff 0.05. In total, 5903 and 183 peaks were identified for *Chd1* KO and WT cells, respectively. After filtering out peaks overlapping with blacklisted regions (http://mitra.stanford.edu/kundaje/akundaje/release/blacklists/mm9-mouse/mm9-blacklist.bed.gz), 5671 and 54 peaks were retained for *Chd1* KO and WT cells, respectively. The identified peaks were associated with genomic annotations using HOMER annotatePeaks.pl[42]. In particular, 1825 peaks in *Chd1* KO cells were mapped around TSSs (−1kb to +100 bp) of 1785 genes (DSB-prone genes).

*Gene annotations.* RefSeq gene annotations were downloaded from UCSC Table Browser. In order to avoid ambiguity, only protein-coding genes with unique TSSs in autosomes were used. Genes whose TSSs ±10 kb overlap with blacklisted regions were further filtered out. As a result, 14,109 genes were retained and used throughout the paper. In Fig. 4B, we further required that the gene has a unique expression value provided by Guzman-Ayala et al.[3] (GSE57609), resulting in 7524 genes. In Fig. 4D–F, 1107 DSB prone genes (intersection between the 1785 genes identified by HOMER and the 14109 filtered genes) were compared with 13002 (=14,109−1107) non-DSB-prone genes. In Fig. 4F, the mean gene length of different isoforms was used for genes with different transcription termination sites (TTS). Gene ontology analysis was performed using the DAVID software[43].

**Reporting summary.** Further information on research design is available in the Nature Research Reporting Summary linked to this article.

## Data availability

The ChIP-seq and DSB-seq data generated in this study have been deposited in the Gene Expression Omnibus (GEO) database under accession code GSE132137. The mass spectrometry data generated in this study have been deposited in the ProteomeXchange Consortium database via the PRIDE[44] partner repository with the dataset identifier PXD024604. The MNase-seq data used in this study are available in the GEO database under the accession code GSE82127. Source data are provided with this paper.

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

## Acknowledgements

We thank members of the Santos Lab for input and critical reading of the manuscript, Richard Lao and members of the UCSF Institute of Human Genetics for assistance with sequencing and sonication. Samples were sequenced at UCSF Institute of Human Genetics Core Facility and Center for Advanced Technology, which is supported by the NIH (5P30CA082103). This research was supported by grants from Bloodwise and Medical Research Council plus the CRUK Manchester Centre award (C5759/A25254) to A.D.W. who is supported by the NIHR Manchester Biomedical Research Centre, 2R01CA163336 to J.S.S., the Sofja Kovalevskaja Award (Humboldt Foundation) to A.B.-K., and by NIH R01GM113014, R01GM123556, CIHR Project Grant 420231, and Canada 150 Research Chair in Developmental Epigenetics to M.R.-S.

## Author contributions

A.B.-K. and M.R.-S. conceived the project. A.B.-K. designed and performed the majority of experiments, with the following exceptions: M.G.-A. isolated, stained, and imaged the embryos, A.J.K.W performed the mass spectrometry runs, H.J. and M.H. performed bioinformatics analyses, under the supervision of J.S.S. A.D.W supervised the mass spectrometry experiments. B.C., Y.K.-K., and M.S. performed follow-up validation experiments, M.R.-S. supervised the project. A.B.-K., and M.R.-S. wrote the manuscript with feedback from all authors.

## Competing interests

The authors declare no competing interests.
