## [Peer Review File · Nature Communications]

Reviewers' comments:

Reviewer #1 (Remarks to the Author):

Bulut-Karslioglu et al describe a new function for the Chd1 nucleosome remodeling protein in repair of double stranded breaks (DSBs). In undamaged ES cells, the authors find that Chd1 co-immunoprecipitates with proteins required for DNA repair and Chd1 promotes recruitment of repair proteins to the rDNA locus. In the absence of Chd1, H2A.X phosphorylation is reduced despite an increase in DSBs near the promoters of active genes, indicating a defect in DNA repair in these cells in the absence of DNA damaging agents. The authors conclude that Chd1 is required for repair of damage associated with active transcription, possibly mediated by Topoisomerase II induced breaks required in some cell types to relieve torsional stress at long, highly expressed genes.

The data are of high quality and the findings are of general interest, but a few points should be fleshed out before publication. In addition, I am confused by their model for how Chd1 simultaneously activates transcription and promotes repair of DSBs, and how these two processes are connected. If transcription stimulates DSB formation, it seems counterintuitive that Chd1 loss reduces transcription while increasing DSB formation. Specific comments and suggestions are listed below.

1. Is transcription initiation required for the increase of promoter-proximal DSBs observed in Chd1 KO cells, despite the fact that transcription is reduced overall in these cells? Although the authors note that DSBs do not correlate with mRNA levels, they do appear to correlate with promoter-proximal Pol II (Fig. 4A), raising the possibility that transcription initiation is necessary for DSBs, even if Pol II pauses after initiating. This could be tested by measuring DSBs after treatment with chemical inhibitors of transcription initiation to prevent any effect on supercoiling caused by Pol II.
2. Do DSBs that accumulate in the absence of Chd1 depend on Topo II activity? Reduced transcription in Chd1 KO cells should lead to reduced torsional strain, which may lead to reduced Topo II activity at these sites. Straightforward inhibition of Topo II activity and measurement of DSBs with and without Chd1 KO should shed light on this issue.
3. A model figure depicting how Chd1-mediated regulation of nucleosome occupancy simultaneously promotes higher levels of transcription and lower levels of DSBs would be helpful. If Chd1-mediated hypertranscription induces torsional strain that is relieved by DSBs and DSB repair also depends on Chd1 through its function to recruit DNA repair proteins, loss of Chd1 would be predicted to have both positive and negative effects on DSBs. On one hand, reduced transcription-induced torsional strain might lead to fewer DSBs as a result of reduced topoisomerase activity. On the other hand, reduced recruitment of repair proteins would be predicted to increase DSBs. Additional discussion of this issue and a model would be helpful, and the above suggestions may shed additional light on this process.

Minor Comments

1. Fig. 1A: Depiction of peptide number (IP-control) seems strange, since larger proteins might be expected to have a larger difference by virtue of their larger number of tryptic fragments. Also, this metric doesn't give an indication of the fold enrichment of each protein in the IP sample.
2. Fig. 1: A browser track of DSB-seq data for the rDNA would be helpful to more precisely visualize the locations of the breaks.
3. Fig. 2A: why is nucleolin staining more intense in Chd1 KO ES cells? Is the nucleolar structure/composition noticeably altered in these cells?

Reviewer #2 (Remarks to the Author):

In the present study, Bulut-Karslioglu et al. report that Chd1 regulates the formation of DNA double-strand breaks at GC-rich promoters of long genes. Although this is a novel and interesting finding, the authors' statement may contain critical issues regarding the interpretation of "DNA repair."

My primary concern is that the current data may not provide evidence on whether Chd1 promotes DSB repair or not. In control cells, DSBs are not detected at rDNA regions; however, gH2AX is

observed. An alternative explanation could be that, although DSBs may not be formed in the control cells, a chromatin structure alteration in the promoters may activate ATM, leading to H2AX phosphorylation. And, in Chd1 KO cells, DSBs are formed without H2AX phosphorylation. Because the authors have not shown that these DSBs are repaired with time, it is unclear whether they are repairable DSBs or not. Or DSBs are derived from cell death-dependent DNA cleavage. Thus, although the authors state that "This work describes a novel role for the chromatin remodeler Chd1 in promoting transcription-mediated DNA break repair" in the first paragraph of the Discussion section; this statement may not be supported by the data of the present study.

1. The authors should confirm whether DSBs are formed in the control cells. DSBs are repaired by either NHEJ, HR, or alt-NHEJ in human cells. Therefore, to determine whether these are repairable DSBs or not, the repair inhibitor or siRNA of DSB factors should be tested.

2. The data show that, in Chd1 KO cells, DSBs accumulate without gH2AX formation. In general, gH2AX is detected in the presence of DSBs. Therefore, in the present study, the authors should validate the DSB assay in Fig. 3 by presenting control data, i.e., the authors should test whether DSB accumulation should be observed in cells defective in a DSB repair factor, such as LIG4 or BRCA2. In addition, DSB formation should be confirmed using an alternative assay, such as the comet assay.

3. Regarding other explanations, Chd1 depletion causes DSB formation, reducing transcription activity. Or Chd1 depletion causes the reduction of transcription activity at the rDNA, forming DSBs. Therefore, to examine whether the reduction of transcription activity generally causes DSBs at the rDNA, the DSB levels should be tested after treatment with Pol1 inhibitor.

Technical concerns:

3. The bands of the IP samples are almost undetectable (Fig. 1D), particularly, for ATM, which presents an undetectable input band. For the main factors, the authors should perform co-IP using the targeted repair proteins, i.e., IP with KAP-1, ATM, PARP-1, or Top2b, and then perform immunoblotting.

4. In Fig 2A, the merge between nucleolin and each repair factor should be shown. Upon merging, pKAP-1 and Top2b may overlap with nucleolin. However, the localization of gH2AX within the nuclei is uncertain.

5. Individual staining images should be shown in Figs. 2D and 2E, and nucleolin should be shown in Fig. 2C. Moreover, the authors' statement in Figs. 2C-E is difficult to evaluate. The authors should validate their statements by providing quantitative values.

6. The referee considers the lack of gH2AX formation in the presence of 5671 DSBs odd. In general, H2AX is phosphorylated by ATM. If H2AX is not truly phosphorylated, then ATM activation (auto-phosphorylation) also does not occur. Therefore, the authors should observe no ATM activation in Chd1 KO cells. Furthermore, to confirm the absence of DSB signaling, the authors should test the formation of 53BP1 foci.

7. The authors have stated that gH2AX is not derived from replication stress. However, in Fig. 3, DSB formation is not detected by the assay in control cells, indicating that gH2AX might be derived from replication stress. The authors should address this issue by performing double staining of gH2AX and BrdU (or EdU).

Reviewer #3 (Remarks to the Author):

In their manuscript "Chd1 regulates repair of promoter-proximal DNA breaks to sustain hypertranscription in embryonic stem cells", Aydan Bulut-Karslioglu and co-authors employed IP-MS and imaging techniques to analyze new functions of chd1 in embryonic stem cells. They found protein interactions to double stranded DNA break repair proteins and validated these findings in several follow up experiments. Overall, the manuscript is well written and the results well-

presented and the conclusions are supported by the data. While the validation part is sound and convincing, the initially carried out IP-MS analysis was done in a very simple and non-statically controlled manner and needs to be improved before publication. Therefore, I recommend publication if the comments below are addressed.

Comments:

Page 11: Please also provide the total number of sequences present in the protein mus musculus database used.

Page 12: Controlling the false-discovery-rate in mass spectrometry-based proteomics experiments is crucial for generating confident results. How was this done for the IP-MS experiment?

Page 12: Please upload all mass spectrometry data used in this manuscript to a public repository like PRIDE (<https://www.ebi.ac.uk/pride/archive/>) or MassIVE <8 <https://massive.ucsd.edu/ProteoSAFe/static/massive.jsp>).

Figure 1A: The information provided is quite limited and needs to be improved in order to understand what data is actually presented. On the y-axis, "Normalized peptide numbers" are presented. First, it is not clear to what these peptide numbers are normalized. Based on the "(IP-Ctrl)", I suppose that actually the peptide number ratio of the IP to the control is presented. Simply taking the ratio of peptide numbers (it is not clear how they were determined nor at what false-discovery-rate) is at most a semi-quantitative approach and only works for strong differences and high peptide spectrum matches (PSMs, which should be rather used). In any case, any quantitative approach requires statistical evaluation to demonstrate the significance of the detected changes. This is totally missing here and needs to be performed. From the Figure 1A, it looks like all identified proteins are considered to be Chd1 interactors, even those with a ratio of 1 being as abundant as in the control (e.g. Polr2h). And why are there no ratios below 1? Were all proteins more abundant and found with more peptides than in the control sample? This is a bit unexpected, since common contaminations would provide higher peptide counts, even at the same concentration, due to the lower complexity of the sample. Was any normalization or filtering applied to the presented data? To sum, as such, the IP-MS data cannot be used. The authors need to perform stringent statically filtering of the quantitative data to provide a confident list of potential protein interactors like provided with the SAINT algorithm (PMID: 21131968). Then only the hits passing the significant thresholds can be used for the GO-enrichment analysis (Figure 1B and the String network analysis (Figure 1C). Also, it is not clear what "bead control" means which was used as a negative control. I would suggest to use the double KO cells as control expressing GFP-Flag.

Figure 1C: Please provide more details to the generated protein network, like the name of the software/server/web page used (Saint) as well as the parameter settings applied, in particular confidence evidence levels.

Figure 1D: The western blots are not of very good quality with the exception of the first two. I do not see any change between WT and control (what

Figure 1: Why was a circular A to D arrangement chosen? To make the graph consistent with the other figures, please place B on the right upper side, C on the lower left and D on the lower right.

We thank the reviewers for their helpful comments and suggestions. We have addressed each of the points raised by the reviewers, as detailed in the point-by-point response below.

Reviewer #1:

Bulut-Karslioglu et al describe a new function for the Chd1 nucleosome remodeling protein in repair of double stranded breaks (DSBs). In undamaged ES cells, the authors find that Chd1 co-immunoprecipitates with proteins required for DNA repair and Chd1 promotes recruitment of repair proteins to the rDNA locus. In the absence of Chd1, H2A.X phosphorylation is reduced despite an increase in DSBs near the promoters of active genes, indicating a defect in DNA repair in these cells in the absence of DNA damaging agents. The authors conclude that Chd1 is required for repair of damage associated with active transcription, possibly mediated by Topoisomerase II induced breaks required in some cell types to relieve torsional stress at long, highly expressed genes.

The data are of high quality and the findings are of general interest, but a few points should be fleshed out before publication. In addition, I am confused by their model for how Chd1 simultaneously activates transcription and promotes repair of DSBs, and how these two processes are connected. If transcription stimulates DSB formation, it seems counterintuitive that Chd1 loss reduces transcription while increasing DSB formation. Specific comments and suggestions are listed below.

1. Is transcription initiation required for the increase of promoter-proximal DSBs observed in Chd1 KO cells, despite the fact that transcription is reduced overall in these cells? Although the authors note that DSBs do not correlate with mRNA levels, they do appear to correlate with promoter-proximal Pol II (Fig. 4A), raising the possibility that transcription initiation is necessary for DSBs, even if Pol II pauses after initiating. This could be tested by measuring DSBs after treatment with chemical inhibitors of transcription initiation to prevent any effect on supercoiling caused by Pol II.

We thank the Reviewer for prompting us to explore further the impact of transcription initiation on DNA break formation in the Chd1 KO ES cells. Considering the fact that Chd1 associates with Pol II and facilitates transcription just downstream of Pol II through pumping DNA towards it¹ Chd1 KO could compromise transcription via accumulation of initiated Pol II at TSSs. As suggested, we inhibited RNA Pol II initiation using triptolide, then quantified DSBs by DSB-qPCR. The results, described in the main text and shown in new Figure 5B, indicate that inhibition of transcription initiation with triptolide does not affect the levels of DSBs at diagnostic promoters in Chd1 KO ES cells, although it does increase breaks in WT ESCs. These data suggest that the DSBs that accumulate in Chd1 KO ES cells are not caused by transcription initiation.

To put these results in context of Chd1-Pol II interaction and localization at TSSs, we plotted Pol II and Chd1 ChIP-seq, and DSB-seq data at TSSs (provided as Reviewer Figure 1 here). DSBs accumulate slightly upstream of Pol II in Chd1 KO cells. While these experiments require further exploration in a future study, taken together with the fact that triptolide treatment does not lead to decreased DSBs in Chd1 KO cells, they results suggest that the breaks already arise at the assembly step. We now expanded on this point in the Results section.

Reviewer Figure 1: Distribution of Chd1, Pol II and DSB signals around TSS. The grey shade shows the nucleosome distribution (data from Voong et al, Cell, 2016¹²). Chd1 peaks downstream of Pol II, while DSBs are enriched slightly upstream of Pol II.

2. Do DSBs that accumulate in the absence of Chd1 depend on Topo II activity? Reduced transcription in Chd1 KO cells should lead to reduced torsional strain, which may lead to reduced Topo II activity at these sites. Straightforward inhibition of Topo II activity and measurement of DSBs with and without Chd1 KO should shed light on this issue.

As suggested by the Reviewer, we performed similar experiments as above in point 1, but using etoposide for Topo II inhibition. We found that etoposide treatment increases DSB formation in both WT and Chd1 KO cells at diagnostic promoters. Thus, inhibition of topoisomerase activity can induce DSBs at active promoters, as has been shown previously², and this is the case even in the absence of Chd1. These results are shown in new Figure 5A, and discussed in the Results section.

3. A model figure depicting how Chd1-mediated regulation of nucleosome occupancy simultaneously promotes higher levels of transcription and lower levels of DSBs would be helpful. If Chd1-mediated hypertranscription induces torsional strain that is relieved by DSBs and DSB repair also depends on Chd1 through its function to recruit DNA repair proteins, loss of Chd1 would be predicted to have both positive and negative effects on DSBs. On one hand, reduced transcription-induced torsional strain might lead to fewer DSBs as a result of reduced topoisomerase activity. On the other hand, reduced recruitment of repair proteins would be predicted to increase DSBs. Additional discussion of this issue and a model would be helpful, and the above suggestions may shed additional light on this process.

We thank the Reviewer for this suggestion, which helped us incorporate the results above with the original results. New Fig 5C now depicts a model whereby hypertranscription-induced transient DSBs are rapidly repaired in the presence of Chd1; when Chd1 is absent, repair of DSBs is deficient, possibly via a combination of reduced recruitment of repair factors and defective +1 nucleosome remodeling, resulting in reduced transcription.

Minor Comments

1. Fig. 1A: Depiction of peptide number (IP-control) seems strange, since larger proteins might be expected to have a larger difference by virtue of their larger number of tryptic fragments. Also, this metric doesn't give an indication of the fold enrichment of each protein in the IP sample.

We thank the Reviewer for pointing this out. Based on the Reviewer's suggestion, we now reanalyzed the IP-MS data and revised Figure 1A-C. The new Figure 1A represents the log₂ fold change (Chd1 IP/negative control IP) on the y axis and the PEP score of the identified proteins in the Chd1 IP sample on the x axis. The PEP score denotes the significance of the recovered hit. We have now performed a more stringent analysis on the identified proteins using the following criteria: Proteins with fewer than 4 peptides in Chd1 IP were filtered out and only those hits passing the log₂FC (Chd1 IP/negative control IP)>2 and PEP score<0.05 were retained for further analysis.

2. Fig. 1: A browser track of DSB-seq data for the rDNA would be helpful to more precisely visualize the locations of the breaks.

This is a valid suggestion, and the DSB-seq data mapped to consensus rDNA is now shown in Supplementary Figure 3A.

3. Fig. 2A: why is nucleolin staining more intense in Chd1 KO ES cells? Is the nucleolar structure/composition noticeably altered in these cells?

While there is variability in the intensity of staining in IF, the Reviewer's observation is overall correct. This is a pattern that we have previously reported, as mentioned in the Results: "We have previously shown that Chd1 binds directly to rDNA and that loss of Chd1 leads to reduced nascent synthesis of rRNA and fragmentation of nucleoli (³ and Fig. 2A)." We have added a note to the Figure 2A legend to further emphasize this point: "Note the fragmentation of nucleoli in Chd1 KO ES cells, as we have previously reported³". This fragmentation of the nucleolus is typically seen with perturbations that lead to reduced rRNA transcription, and may appear as a more intense/dispersed Nucleolin staining.

Reviewer #2:

In the present study, Bulut-Karslioglu et al. report that Chd1 regulates the formation of DNA double-strand breaks at GC-rich promoters of long genes. Although this is a novel and interesting finding, the authors' statement may contain critical issues regarding the interpretation of "DNA repair."

My primary concern is that the current data may not provide evidence on whether Chd1 promotes DSB repair or not. In control cells, DSBs are not detected at rDNA regions; however, γH2AX is observed. An alternative explanation could be that, although DSBs may not be formed in the control cells, a chromatin structure alteration in the promoters may activate ATM, leading to H2AX phosphorylation. And, in Chd1 KO cells, DSBs are formed without H2AX phosphorylation. Because the authors have not shown that these DSBs are repaired with time, it is unclear whether they are repairable DSBs or not. Or DSBs are derived from cell death-dependent DNA cleavage. Thus, although the authors state that "This work describes a novel role for the chromatin remodeler Chd1 in promoting transcription-mediated DNA break repair" in the first paragraph of the Discussion section; this statement may not be supported by the data of the present study.

1. The authors should confirm whether DSBs are formed in the control cells. DSBs are repaired by either NHEJ, HR, or alt-NHEJ in human cells. Therefore, to determine whether these are repairable DSBs or not, the repair inhibitor or siRNA of DSB factors should be tested.

We are grateful to the Reviewer for prompting us to clarify the nature of DSBs in wild-type cells. We note that promoter-proximal DSBs have already been reported in untreated wild-type cells⁴⁻⁸. In agreement, we detect a baseline level of DSBs in wild-type ES cells (Fig. 4A), and this is not a novel aspect of our study. These DSBs are sharply increased in Chd1 KO ES cells, while

gH2AX is decreased, suggestive of defective repair. As suggested by the Reviewer, we performed RNAi for Atm or Brca2, as well as inhibition of Atm using the small molecule KU-60019. The data, shown here as Reviewer Figure 2, reveal that either treatment at several timepoints leads to a decrease of DSBs in WT cells. While the reasons for this remain at present unknown and warrant future study, the consistency of these results suggests that they may be due to secondary effects of the treatments, such as potential reductions in transcription. Overall, the reductions in DSBs observed with these treatments further support the notion that DSBs do form in control, wild-type ES cells, which was the concern of the Reviewer.

Reviewer Figure 2: Status of DSBs upon perturbation of Atm and Brca2 function. A) Cells were treated with an ATM inhibitor for 20 min and DSBs at rDNA loci were quantified by DSB-qPCR. B) Atm and Brca2 were knock-down using siRNAs and DSBs at rDNA loci were quantified using DSB-qPCR. Fold change relative to control siRNA knockdown are shown.

2. The data show that, in Chd1 KO cells, DSBs accumulate without gH2AX formation. In general, gH2AX is detected in the presence of DSBs. Therefore, in the present study, the authors should validate the DSB assay in Fig. 3 by presenting control data, i.e., the authors should test whether DSB accumulation should be observed in cells defective in a DSB repair factor, such as LIG4 or BRCA2. In addition, DSB formation should be confirmed using an alternative assay, such as the comet assay.

Please see point 1 above for a more detailed response including our data with Atm RNAi, Brca1 RNAi and chemical inhibition of ATM (Reviewer Figure 2). We focused on using DSB-qPCR to further explore the results of the original submission. While comet assays are often used for studies of DNA damage exogenously induced by chemical agents, they would not give us the resolution to assess specific genomic locations found to be vulnerable to Chd1 loss in undamaged cells. To validate the DSB-qPCR approach, we now include data showing that inhibition of Topo II with Etoposide reliably induces DSBs at the diagnostic regions used (new Figure 5A).

3. Regarding other explanations, Chd1 depletion causes DSB formation, reducing transcription activity. Or Chd1 depletion causes the reduction of transcription activity at the rDNA, forming DSBs. Therefore, to examine whether the reduction of transcription activity generally causes DSBs at the rDNA, the DSB levels should be tested after treatment with Pol1 inhibitor.

We thank the Reviewer for this very useful suggestion. It is true that Chd1 KO does lead to reduced activity of RNA Pol I, as we have shown in ref. 3 and is further validated here in Fig. 3D. We now include new data showing that treatment of WT ES cells with RNA Pol I inhibitor CX-5461 does not induce DSB formation at rDNA (new Figure 3E), contrary to what is found in Chd1 KO ES cells. Hence, these data indicate that DSB formation at rDNA in Chd1 KO ES cells is not an indirect consequence of reduced Pol I transcription, further supporting the model of defective repair in the mutant cells.

Technical concerns:

3. The bands of the IP samples are almost undetectable (Fig. 1D), particularly, for ATM, which presents a undetectable input band. For the main factors, the authors should perform co-IP using the targeted repair proteins, i.e., IP with KAP-1, ATM, PARP-1, or Top2b, and then perform immunoblotting.

Most if not all of the identified interactors of Chd1 are sub-stoichiometric and therefore are less visible compared to the input bands on the same blot. In the case of phosphorylated Atm (p-Atm), this is the activated version of the protein and is present at very low levels. It is not visible in the input lane containing all nuclear proteins, but is recovered in the Chd1 IP. Nevertheless, we now repeated several of these IP-WB results and included these new biological replicates in Figure S1C. These new Co-IP WB results further support the reproducibility of the interactions.

4. In Fig 2A, the merge between nucleolin and each repair factor should be shown. Upon merging, pKAP-1 and Top2b may overlap with nucleolin. However, the localization of gH2AX within the nuclei is uncertain.

We agree with the Reviewer's suggestion. We revised Fig. 2A so as to only include antibody combinations that allow a clear molecular demarcation of the nucleolus. Revised Fig. 2A now clearly documents a strong nucleolar accumulation of Atm and Top2b, which is reproducible. Our additional experiments with different antibodies for pKap1 and gH2AX and using different fixations methods overall revealed a broad staining for these two proteins in foci throughout the nucleus, including the nucleolus, but with no consistent enrichment in the nucleolus. Given that these IF images were not particularly informative, we decided to not include them, focusing instead on the ChIP-qPCR data for pKap1 and gH2AX, which show a quantitative decrease in these two factors at rDNA in Chd1 KO ES cells. The Results text has been revised accordingly.

5. Individual staining images should be shown in Figs. 2D and 2E, and nucleolin should be shown in Fig. 2C. Moreover, the authors' statement in Figs. 2C-E is difficult to evaluate. The authors should validate their statements by providing quantitative values.

We are currently not in a position to add new images on the mouse embryo data and respectfully believe that they are not necessary - the data are indeed qualitative, not quantitative, since both points made in 2C-E are immediately apparent from the images: i) gH2AX clearly co-localizes with the nucleolus (overlap gH2AX/nucleolin in D, exclusion between gH2AX and both DAPI and S2p RNA Pol II in 2E), and ii) gH2AX is entirely absent in Chd1 KO embryos (Fig. 2C and E). To clarify the numbers of embryos used, we have added the following statement to Fig. 2 legend: "A minimum of 4 embryos were analyzed per genotype per stage."

6. The referee considers the lack of gH2AX formation in the presence of 5671 DSBs odd. In general, H2AX is phosphorylated by ATM. If H2AX is not truly phosphorylated, then ATM

activation (auto-phosphorylation) also does not occur. Therefore, the authors should observe no ATM activation in Chd1 KO cells. Furthermore, to confirm the absence of DSB signaling, the authors should test the formation of 53BP1 foci.

The Reviewer is correct to point out the intriguing fact that gH2AX is reduced in the context of increased DSBs in Chd1 KO cells. This result supports the notion that the repair of DSBs induced at the promoters of transcribed genes, including rDNA, is defective in Chd1 KO cells. We have followed up on the Reviewer's suggestion to analyze pATM levels by western blotting. The data, added to Supplementary Figure 1D, shows that pATM is detected at similar levels in both control and Chd1 KO ES cells. This is perhaps not too surprising, given that western blotting measures overall global cellular levels of pATM, and that gH2AX is still detected in Chd1 KO ES cells, albeit at somewhat reduced levels.

One possible explanation for the disconnect between increased DNA breaks and decreased gH2AX is defective gH2AX spreading due to compromised H2A.X deposition in Chd1 KO cells. H2A.X is deposited by the FACT complex⁹. The interaction between the FACT complex and Chd1 was previously shown. Recently, it has been proposed that FACT spreads in a Chd1-dependent manner¹⁰. Therefore it is conceivable that H2A.X deposition, and as a result, H2A.X phosphorylation and amplification of the DNA damage signal, could be compromised in Chd1 KO cells. We have expanded on this point now in the Discussion section.

Regarding 53BP1 foci, they are used as a marker of DNA breaks induced by exogenous damage agents, which is not the case here. Nevertheless, we performed 53BP1 IF and can detect a few foci at similar levels in control and KO ES cells. Positive control WT cells treated with Aphidicolin show robust induction of 53BP1 foci. Both of these pieces of data are now shown in new Supplementary Figure 2C. We have therefore added the following statement to the Results section: "A few foci of 53BP1, a marker of DNA breaks induced by exogenous damage agents, are detected at similar levels in both WT and Chd1 KO ES cells, although they are highly increased with Aphidicolin treatment (Fig. S2C). These findings suggest that aspects of the mechanism of repair of endogenous, transcription-induced DNA breaks are distinct from repair of DSBs induced by exogenous agents."

7. The authors have stated that gH2AX is not derived from replication stress. However, in Fig. 3, DSB formation is not detected by the assay in control cells, indicating that gH2AX might be derived from replication stress. The authors should address this issue by performing double staining of gH2AX and BrdU (or EdU).

We thank the Reviewer for prompting us to investigate the relationship between gH2AX and the replication. Our speculation in the Discussion about replicative stress stemmed from the fact that gH2AX was detected in IF across all ES and epiblast cells. We agree that this point could be explored much better with gH2AX/EdU flow cytometry. We now include new flow cytometry data showing that gH2AX staining is not limited to S phase cells, as might be expected from replication stress, but is found across all stages of the cell cycle, in both control and Chd1 KO ES cells. These data, shown in new Supplementary Figure 4 and now referred to in the Discussion, support our original speculation that gH2AX accumulation in ES cells is not strictly due to replication stress.

Reviewer #3:

In their manuscript "Chd1 regulates repair of promoter-proximal DNA breaks to sustain hypertranscription in embryonic stem cells", Aydan Bulut-Karslioglu and co-authors employed IP-MS and imaging techniques to analyze new functions of chd1 in embryonic stem cells. They found protein interactions to double stranded DNA break repair proteins and validated these

findings in several follow up experiments. Overall, the manuscript is well written and the results well-presented and the conclusions are supported by the data. While the validation part is sound and convincing, the initially carried out IP-MS analysis was done in a very simple and non-statically controlled manner and needs to be improved before publication. Therefore, I recommend publication if the comments below are addressed.

Page 11: Please also provide the total number of sequences present in the protein mus musculus database used.

We thank the Reviewer for helping us present the mass spectrometry data in a clearer manner, in this point and below. The total number of sequences present in the database used is 16403. This information is now also entered in the Experimental Procedures section.

Page 12: Controlling the false-discovery-rate in mass spectrometry-based proteomics experiments is crucial for generating confident results. How was this done for the IP-MS experiment?

The false discovery rate (FDR) of protein identification was controlled using a target-decoy searching strategy¹¹. The maximum allowed FDR for protein identification was set to 1%. This is now stated in the same Experimental Procedures section.

Page 12: Please upload all mass spectrometry data used in this manuscript to a public repository like PRIDE (<https://www.ebi.ac.uk/pride/archive/>) or MassIVE (<[8https://massive.ucsd.edu/ProteoSAFe/static/massive.jsp](https://massive.ucsd.edu/ProteoSAFe/static/massive.jsp)>).

The mass spectrometry data has now been deposited in the PRIDE repository under accession number PXD024604. This number is also now included in the manuscript under Data Availability. The reviewers can access the data using the following account details:

Username: reviewer_pxd024604@ebi.ac.uk

Password: BCTn4NUq

Figure 1A: The information provided is quite limited and needs to be improved in order to understand what data is actually presented. On the y-axis, "Normalized peptide numbers" are presented. First, it is not clear to what these peptide numbers are normalized. Based on the "(IP-Ctrl)", I suppose that actually the peptide number ratio of the IP to the control is presented. Simply taking the ratio of peptide numbers (it is not clear how they were determined nor at what false-discovery-rate) is at most a semi-quantitative approach and only works for strong differences and high peptide spectrum matches (PSMs, which should be rather used). In any case, any quantitative approach requires statistical evaluation to demonstrate the significance of the detected changes. This is totally missing here and needs to be performed. From the Figure 1A, it looks like all identified proteins are considered to be Chd1 interactors, even those with a ratio of 1 being as abundant as in the control (e.g. Polr2h). And why are there no ratios below 1? Were all proteins more abundant and found with more peptides than in the control sample? This is a bit unexpected, since common contaminations would provide higher peptide counts, even at the same concentration, due to the lower complexity of the sample. Was any normalization or filtering applied to the presented data? To sum, as such, the IP-MS data cannot be used. The authors need to perform stringent statically filtering of the quantitative data to provide a confident list of potential protein interactors like provided with the SAINT algorithm (PMID: 21131968). Then only the hits passing the significant thresholds can be used for the GO-enrichment analysis (Figure 1B and the String network analysis (Figure 1C). Also, it is not clear what "bead control" means which was used as a negative control. I would suggest to use the double KO cells as control expressing GFP-Flag. We thank the Reviewer for prompting us to better analyze and display the IP-MS data. Based on the Reviewer's suggestion, we now reanalyzed the IP-MS data and revised Figure 1A-C. The

new Figure 1A represents the log₂ fold change (Chd1 IP/negative control IP) on the y axis and the PEP score of the identified proteins in the Chd1 IP sample on the x axis. The PEP score denotes the significance of the recovered hit. We have now performed a more stringent analysis on the identified proteins using the following criteria: Proteins with fewer than 4 peptides in Chd1 IP were filtered out and only those hits passing the log₂FC (Chd1 IP/negative control IP)>2 and PEP score<0.05 were retained for further analysis. Negative control used in this experiment is IgG-based IP. In the text, we have now revised this nomenclature as 'IgG control' to make it more clear. The legend of Figure 1 and the relevant Experimental Procedures section were revised accordingly.

Figure 1C: Please provide more details to the generated protein network, like the name of the software/server/web page used (Saint) as well as the parameter settings applied, in particular confidence evidence levels.

To generate the protein network, we used the STRING database with the following parameters: Network type: Full STRING network, meaning of network edges: confidence (line thickness indicates the strength of data support), active interaction sources: experiments, textmining, databases, co-expression, neighborhood, gene fusion, co-occurrence, minimum required interaction score=0.4. These details have now been added to the Experimental Procedures section. The GO pathway analysis and protein network analysis have now been updated using the stringently filtered hits (see above).

Figure 1D: The western blots are not of very good quality with the exception of the first two. I do not see any change between WT and control

Most if not all of the identified interactors of Chd1 are sub-stoichiometric and therefore are less visible compared to the input bands on the same blot. In the case of phosphorylated Atm (p-Atm), this is the activated version of the protein and is present at very low levels. It is not visible in the input lane containing all nuclear proteins, but is recovered in the Chd1 IP. Nevertheless, we now provide additional IP-WB results in new biological replicates in Figure S1. Please see also the response above to technical comment #3 of Reviewer 1.

Figure 1: Why was a circular A to D arrangement chosen? To make the graph consistent with the other figures, please place B on the right upper side, C on the lower left and D on the lower right.

The circular arrangement was chosen to be able to show the protein network data next to its corresponding GO term. We have now made the requested change in Figure 1.

1. Farnung, L., Ochmann, M., Engholm, M. & Cramer, P. Structural basis of nucleosome transcription mediated by Chd1 and FACT. *Biorxiv* 2020.11.30.403857 (2020)
doi:10.1101/2020.11.30.403857.

2. Baranello, L. *et al.* DNA break mapping reveals topoisomerase II activity genome-wide. *International journal of molecular sciences* **15**, 13111–13122 (2014).

3. Guzman-Ayala, M. *et al.* Chd1 is essential for the high transcriptional output and rapid growth of the mouse epiblast. - PubMed - NCBI. *Development* **142**, 118–127 (2014).

4. Williamson, L. M. & Lees-Miller, S. P. Estrogen receptor α -mediated transcription induces cell cycle-dependent DNA double-strand breaks. *Carcinogenesis* **32**, 279–285 (2011).

5. Ju, B. G. *et al.* A topoisomerase IIbeta-mediated dsDNA break required for regulated transcription. *Science (New York, N.Y.)* **312**, 1798–1802 (2006).
6. Wong, M. M., Belew, M. D., Kwieraga, A., Nhan, J. D. & Michael, W. M. Programmed DNA Breaks Activate the Germline Genome in *Caenorhabditis elegans*. *Developmental Cell* **46**, 302-315.e5 (2018).
7. Bunch, H. *et al.* Transcriptional elongation requires DNA break-induced signalling. *Nature Communications* **6**, 10191 (2015).
8. Endres, T. *et al.* Ubiquitylation of MYC couples transcription elongation with double-strand break repair at active promoters. *Mol Cell* **81**, 830-844.e13 (2021).
9. Piquet, S. *et al.* The Histone Chaperone FACT Coordinates H2A.X-Dependent Signaling and Repair of DNA Damage. *Mol Cell* **72**, 888-901.e7 (2018).
10. Jeronimo, C. *et al.* FACT is recruited to the +1 nucleosome of transcribed genes and spreads in a Chd1-dependent manner. *Biorxiv* 2020.08.20.259960 (2020)
doi:10.1101/2020.08.20.259960.
11. Elias, J. E. & Gygi, S. P. Target-decoy search strategy for increased confidence in large-scale protein identifications by mass spectrometry. *Nat Methods* **4**, 207–214 (2007).
12. Voong, L. N. *et al.* Insights into Nucleosome Organization in Mouse Embryonic Stem Cells through Chemical Mapping. *Cell* **167**, 1555-1570.e15 (2016).

REVIEWER COMMENTS

Reviewer #1 (Remarks to the Author):

The authors have largely addressed my concerns, but I have one remaining question about their conclusions. The authors previously showed that loss of Chd1 broadly reduces transcription, and here show that Chd1 loss leads to increased DSBs. To test whether the reduction in transcription may be the cause of the increase in DSBs, triptolide is used to block transcription initiation, and a similar increase in DSBs was observed in triptolide-treated cells that was observed when Chd1 is deleted. When the two factors that increase DSBs (triptolide and Chd1 loss) were combined, there was no additional increase in DSBs.

From these data, the authors propose “the accumulation of unrepaired DNA breaks in Chd1 KO cells compromises nascent transcriptional output and leads to the proliferation defects and ultimate developmental arrest of the rapidly expanding post-implantation epiblast.” In my opinion, the likely cause and effect are reversed. Since triptolide causes an increase in DSBs (just as Chd1 deletion does), the most straightforward explanation, in my view, is that loss of transcription increases DSBs. In this scenario, it would follow that deletion of Chd1, which causes a reduction in global transcription, would also increase DSBs. The authors point out that when the two inducers of DSBs (triptolide and Chd1 deletion) are combined there is no further increase in DSBs over Chd1 deletion alone. Perhaps this is because DNA damage is increased when transcription is below some threshold (which may vary by gene), and further reductions in transcription have no additional effect on DSBs. If either triptolide or Chd1 “saturates” the increased DNA damage near gene promoters, combination of the two would presumably not increase damage beyond either individual treatment. On the other hand, the RNA Polymerase I inhibitor appears to show only modest, non-statistically significant increases in DSBs. It may be that the rDNA and coding genes behave differently.

The effects of triptolide or Chd1 deletion are partly obscured by plotting the DSBs resulting from each on separate graphs in Fig. 5B, preventing the increases in DSBs at each gene resulting from triptolide, Chd1 KO, or both from being compared directly. I suggest the authors plot these data in the same graph, where WT/no triptolide is set to 1 for each gene, to compare the individual and combined effects of triptolide and Chd1 KO more directly. (A similar argument could be made for Fig. 5A.)

In addition, the authors should add an explanation somewhere in the manuscript of the reasoning behind their model that increased DSBs cause reduced transcription in Chd1 KO cells, rather than vice versa.

Minor comment:

Figure 1: PEP score needs to be defined in the methods or figure legend.

Reviewer #2 (Remarks to the Author):

The manuscript has been improved; however, unfortunately, my concerns have not been fully addressed. The major issue is regarding the statement, “Chd1 promotes the repair of DSBs,” which is not fully supported by the current data. The authors insisted that their data showed the accumulation of DSBs in Chd1 without exogenous DNA damage, which suggested that Chd1 played a role in DSB repair. However, the authors are likely to obtain the same result if Chd1 suppressed the occurrence of DSB induction; for example, by alternating chromatin compaction (e.g., PMID: 32710624) or enhancing Topo2 molecules at promoter regions. In this scenario, Chd1 does not need to play a functional role in repair. I agree with the findings of accumulation of DSBs in Chd1 cells; however, I am not convinced by the conclusion made by the authors suggesting that Chd1 promotes the repair of DSBs. The authors should reconsider their conclusion or verify that Chd1 promotes DSB repair. If the authors choose the latter, I suggest the below experiments. The authors found several repair proteins that interacted with Chd1. PARP is involved in BER and alt-NHEJ. ATM is a signaling factor that is mainly activated at DSB ends. Despite the critical role of

ATM in signaling, it is involved in only a proportion of DSB repair (approximately 15%–20% of total; PMID: 21642969 and 18657500). As shown in these previous studies, ATM-dependent KAP-1 phosphorylation promotes DSB repair but only in approximately 15%–20% of total DSBs. Such DSB fractions seem to be associated with heterochromatin. More importantly, in this case, pATM-KAP-1 promotes NHEJ. In the Reviewer Figure 2 of the rebuttal letter, the authors tested the ATM inhibitors, siAtm and siBrca2. However, it is not clear why the authors did not test the NHEJ inhibitor. NHEJ is a major DSB repair pathway throughout cell cycle phases. If NHEJ is not required for repair, the authors should also test the PARP inhibitor, which blocks alt-NHEJ. Although the authors do not need to test whether ATM-KAP-1 axis is important in Chd1-dependent repair, KAP-1 depletion or KAP-1 phosphor-mutant should reveal the repair defect. If the authors fail to show the repair defect in these repair-defective backgrounds, this would raise the question of why it is only the Chd1 KO cells that show the repair defect. It would be more reasonable to suggest that greater DSBs in Chd1 KO cells are caused by high DSB induction, rather than the repair defect. I am not against the authors stating that, for example, Chd1 protects genome integrity at promoter regions by preventing DSB formation.

Furthermore, if the authors insist on making their original claim, I have a technical concern about their IP experiment. Although the authors provided additional data in Supplementary Figure 1, the quality of the western blot and immunoprecipitation is still poor. The authors showed a very broad band using the p-ATM antibody after Flag-IP. This type of signal can also be detected by isotype antibodies. Furthermore, in the input of Supplementary Figure 1C, there are no ATM and p-ATM bands. Thus, it is unclear how the authors can demonstrate that the IP bands were derived from p-ATM and ATM. Indeed, the band exists in the input of the KI lane (Supplementary Figure 1C) but it is not clear if this is ATM. The size of this signal differs considerably from the band of the IP lane. Although the authors may argue that all the interactors were not sub-stoichiometric compared with the input, the authors should perform an IP-western blot for each target and adjust the percentage of the input. In addition, the authors should perform isotype control (at least for p-ATM and ATM) and show the percentage of the input for all IP experiments. Given that the role of Chd1 in the repair of DSBs is not convincing, evidence of an interaction between Chd1 and known repair factors is critical. In addition, there are multiple bands in the p-ATM lane. I understand that the p-ATM signal should be very weak without exogenous DNA damage. Nevertheless, to clarify the p-ATM signal, the authors should show the p-ATM band by treatment with ETP or other exogenous DNA damage.

Reviewer #3 (Remarks to the Author):

The authors have considerably improved the manuscript and addressed all comments satisfactorily. It is now acceptable for publication.

We thank the Reviewers for their continued constructive feedback on our manuscript. We address each of their points below.

Reviewer #1 (Remarks to the Author):

The authors have largely addressed my concerns, but I have one remaining question about their conclusions. The authors previously showed that loss of Chd1 broadly reduces transcription, and here show that Chd1 loss leads to increased DSBs. To test whether the reduction in transcription may be the cause of the increase in DSBs, triptolide is used to block transcription initiation, and a similar increase in DSBs was observed in triptolide-treated cells that was observed when Chd1 is deleted. When the two factors that increase DSBs (triptolide and Chd1 loss) were combined, there was no additional increase in DSBs.

From these data, the authors propose “the accumulation of unrepaired DNA breaks in Chd1 KO cells compromises nascent transcriptional output and leads to the proliferation defects and ultimate developmental arrest of the rapidly expanding post-implantation epiblast.” In my opinion, the likely cause and effect are reversed. Since triptolide causes an increase in DSBs (just as Chd1 deletion does), the most straightforward explanation, in my view, is that loss of transcription increases DSBs. In this scenario, it would follow that deletion of Chd1, which causes a reduction in global transcription, would also increase DSBs. The authors point out that when the two inducers of DSBs (triptolide and Chd1 deletion) are combined there is no further increase in DSBs over Chd1 deletion alone. Perhaps this is because DNA damage is increased when transcription is below some threshold (which may vary by gene), and further reductions in transcription have no additional effect on DSBs. If either triptolide or Chd1 “saturates” the increased DNA damage near gene promoters, combination of the two would presumably not increase damage beyond either individual treatment. On the other hand, the RNA Polymerase I inhibitor appears to show only modest, non-statistically significant increases in DSBs. It may be that the rDNA and coding genes behave differently.

While it is true that triptolide treatment is used to block transcription initiation, resulting in reduced nascent transcription and increased DSBs, we would still argue that reduced transcription is unlikely to be the primary cause of increased DSBs in the Chd1 KO ES cells. First, and as we noted, the propensity to accumulate DSBs in Chd1 KO ES cells does not correlate with wild-type gene expression levels (Fig. 4B-D, Spearman $\rho = 0.012$, $p > 0.306$) or reduced expression upon Chd1 loss¹ (Fig. S3B). The Reviewer correctly pointed out that there is a correlation of DSBs with the abundance of promoter-proximal Pol II in wt cells. However, comparison of 1st and 2nd quartiles in Fig. 4A shows a significant difference in DSBs but a minor decrease in Pol II abundance, arguing against the notion that DSBs happen as a direct function of Pol II occupancy.

Second, we do not detect abundant DSBs in wt cells. Given that there are numerous genes which are expressed at high, moderate, or low levels, for which we do not detect major differences in DSBs, we would argue that it is unlikely that transcription levels are the primary determinant of DSB formation.

Third, although triptolide treatment reduces nascent transcription by blocking Pol II initiation, the results are confounded by potential secondary effects. We have previously shown that ES euchromatin and transcriptional landscapes are acutely controlled by prolific transcription and translation (Figure 5 in Bulut-Karslioglu, Macrae et al., Cell Stem Cell, 2018²). Euchromatin regulators including Chd1 and Pol II are unstable proteins and are quickly depleted in the absence of constant translation (significant depletion is detectable by 1h, Figure 5B, Bulut-Karslioglu, Macrae et al., Cell Stem Cell, 2018). The mRNAs of chromatin and translational regulators are also unstable³. Although we performed triptolide treatment for 1 hour to avoid secondary effects, it is still possible that the treatment results in an altered chromatin landscape, including the depletion of Chd1, which would then result in increased DNA breaks.

Fourth, triptolide itself is not expected to be a direct mimic of the reduction in nascent transcription in Chd1 KO. Prolonged culture of ESCs in triptolide would not be possible, whereas we can continuously propagate Chd1 KO ESCs, with only a mild-to-moderate self-renewal deficit¹. Further complicating

matters, triptolide has also been shown to induce DNA breaks and proposed to act in this manner independently of effects on transcription, including via inhibiting DNA-PKcs^{4,5}.

A more detailed investigation of Pol II and nascent transcription dynamics in Chd1 KO cells is necessary to reveal the dependence of DSBs on Pol II initiation or transcription in Chd1 KO cells, however we respectfully deem this out of scope of this work. Nevertheless, the comments of the Reviewer are valid and may occur to readers as well, and therefore have more thoroughly discussed these points in the text. The new text added to each section are as follows:

Results, page 6 (new text is underlined):

“Although DSB occurrence in Chd1 KO cells does not correlate with gene expression levels in wt cells or with differential expression in Chd1 KO cells, DSBs do correlate to some extent with Pol II occupancy at promoters (Fig. 4A). We note that this correlation is not linear, as the 1st and 2nd quartiles of highest DSB-accumulating genes have a greater difference in DSB levels than Pol II occupancy in control cells. Considering the fact that Chd1 associates with Pol II and facilitates transcription just downstream of Pol II through pumping DNA towards it⁶, Chd1 KO could compromise transcription via accumulation of initiated Pol II at TSSs. To probe the role of transcription initiation on DSB occurrence, we tested DSB levels upon inhibition of transcription initiation via triptolide treatment (Fig. 5B). Inhibition of transcription initiation does increase DSBs in wt cells to levels comparable to Chd1 KO cells but does not further increase DSBs in Chd1 KO cells. These data suggest that DSBs that form in Chd1 KO ES cells may occur during Pol II loading and assembly, while reduced nascent transcription may further contribute to DSBs accumulation.”

Discussion, page 7 (added text):

“Although we did not observe a direct correlation between DSB formation and expression levels in wt or expression changes in Chd1 KO cells, a role for reduced transcriptional output on DSB formation via Pol II accumulation or pausing is a possibility that warrants further studies.”

The effects of triptolide or Chd1 deletion are partly obscured by plotting the DSBs resulting from each on separate graphs in Fig. 5B, preventing the increases in DSBs at each gene resulting from triptolide, Chd1 KO, or both from being compared directly. I suggest the authors plot these data in the same graph, where WT/no triptolide is set to 1 for each gene, to compare the individual and combined effects of triptolide and Chd1 KO more directly. (A similar argument could be made for Fig. 5A.)

As the Reviewer suggested, we now present combined panels in Figures S4A and B to allow direct comparison of the effect of triptolite or etoposide treatment on the wt and Chd1 KO cells.

In addition, the authors should add an explanation somewhere in the manuscript of the reasoning behind their model that increased DSBs cause reduced transcription in Chd1 KO cells, rather than vice versa.

We more thoroughly discussed the model now in the text as explained in point 1 above.

Minor comment:

Figure 1: PEP score needs to be defined in the methods or figure legend.

We have now defined the PEP score in the methods and legend as follows: The posterior error probability (PEP) score denotes the probability that the identified peptide is correct.

The PEP score calculation follows standard methods and therefore we did not provide further details.

Reviewer #2 (Remarks to the Author):

The manuscript has been improved; however, unfortunately, my concerns have not been fully addressed. The major issue is regarding the statement, “Chd1 promotes the repair of DSBs,” which is not fully supported by the current data. The authors insisted that their data showed the accumulation of DSBs in Chd1 without exogenous DNA damage, which suggested that Chd1 played a role in DSB

repair. However, the authors are likely to obtain the same result if Chd1 suppressed the occurrence of DSB induction; for example, by alternating chromatin compaction (e.g., PMID: 32710624) or enhancing Topo2 molecules at promoter regions. In this scenario, Chd1 does not need to play a functional role in repair. I agree with the findings of accumulation of DSBs in Chd1 cells; however, I am not convinced by the conclusion made by the authors suggesting that Chd1 promotes the repair of DSBs. The authors should reconsider their conclusion or verify that Chd1 promotes DSB repair. If the authors choose the latter, I suggest the below experiments.

The authors found several repair proteins that interacted with Chd1. PARP is involved in BER and alt-NHEJ. ATM is a signaling factor that is mainly activated at DSB ends. Despite the critical role of ATM in signaling, it is involved in only a proportion of DSB repair (approximately 15%–20% of total; PMID: 21642969 and 18657500). As shown in these previous studies, ATM-dependent KAP-1 phosphorylation promotes DSB repair but only in approximately 15%–20% of total DSBs. Such DSB fractions seem to be associated with heterochromatin. More importantly, in this case, pATM-KAP-1 promotes NHEJ. In the Reviewer Figure 2 of the rebuttal letter, the authors tested the ATM inhibitors, siAtm and siBrca2. However, it is not clear why the authors did not test the NHEJ inhibitor. NHEJ is a major DSB repair pathway throughout cell cycle phases. If NHEJ is not required for repair, the authors should also test the PARP inhibitor, which blocks alt-NHEJ. Although the authors do not need to test whether ATM-KAP-1 axis is important in Chd1-dependent repair, KAP-1 depletion or KAP-1 phosphor-mutant should reveal the repair defect. If the authors fail to show the repair defect in these repair-defective backgrounds, this would raise the question of why it is only the Chd1 KO cells that show the repair defect. It would be more reasonable to suggest that greater DSBs in Chd1 KO cells are caused by high DSB induction, rather than the repair defect. I am not against the authors stating that, for example, Chd1 protects genome integrity at promoter regions by preventing DSB formation.

Furthermore, if the authors insist on making their original claim, I have a technical concern about their IP experiment. Although the authors provided additional data in Supplementary Figure 1, the quality of the western blot and immunoprecipitation is still poor. The authors showed a very broad band using the p-ATM antibody after Flag-IP. This type of signal can also be detected by isotype antibodies. Furthermore, in the input of Supplementary Figure 1C, there are no ATM and p-ATM bands. Thus, it is unclear how the authors can demonstrate that the IP bands were derived from p-ATM and ATM. Indeed, the band exists in the input of the KI lane (Supplementary Figure 1C) but it is not clear if this is ATM. The size of this signal differs considerably from the band of the IP lane. Although the authors may argue that all the interactors were not sub-stoichiometric compared with the input, the authors should perform an IP-western blot for each target and adjust the percentage of the input. In addition, the authors should perform isotype control (at least for p-ATM and ATM) and show the percentage of the input for all IP experiments. Given that the role of Chd1 in the repair of DSBs is not convincing, evidence of an interaction between Chd1 and known repair factors is critical. In addition, there are multiple bands in the p-ATM lane. I understand that the p-ATM signal should be very weak without exogenous DNA damage. Nevertheless, to clarify the p-ATM signal, the authors should show the p-ATM band by treatment with ETP or other exogenous DNA damage. We thank the Reviewer for their comments. We realize that we may have used the phrase “promotes DNA repair” in too loose a fashion, implying a direct biochemical role for Chd1 in activating specific molecular steps of repair of DSBs. The Reviewer is correct that, as we previously implied in the Discussion, more work is required to dissect the precise biochemical mechanisms by which Chd1 promotes genomic integrity in the context of hypertranscription. For example, it is possible that Chd1 facilitates the religation activity of Top2b, and in parallel recruits DNA repair factors as a back-up mechanism that may be particularly important in hypertranscribing cells. We consider the comments of the Reviewer justifiable and agree that it is reasonable to discuss alternative explanations in the manuscript. Following directly on the suggestions of the Reviewer, the revisions are as follows:

Title:

Revised from “Chd1 regulates repair of promoter-proximal DNA breaks to sustain hypertranscription in embryonic stem cells” to “Chd1 protects genome integrity at promoters to sustain hypertranscription in embryonic stem cells”.

Results, page 6 (new text is underlined):

“Finally, we explored the relationship between DSBs induced by Chd1 loss in ES cells and the activities of Topoisomerase, transcription initiation and DNA repair. Topoisomerase creates then ligates DSBs, unless the cells are treated with etoposide which blocks its ligation activity. Treatment of control cells with etoposide increases DSB formation, indicating that the DSBs do occur in wt cells but are promptly repaired by Topoisomerases in the presence of Chd1 (Fig. 5A). Etoposide treatment further increases DSB levels in Chd1 KO cells, suggesting that the DSBs observed in Chd1 KO cells are not simply due to defective Topoisomerase activity, a finding that requires further investigation. Overall, inhibition of topoisomerase activity can induce DSBs at active promoters, as has been shown previously⁷, and this is the case even in the absence of Chd1.”

Discussion, opening sentence, page 7:

“This work describes a novel role for the chromatin remodeler Chd1 in protecting genome integrity at promoter regions by preventing DSB accumulation in pluripotent stem cells.”

Discussion, page 7 (added text):

“Similarly, understanding to what extent Chd1 specifically suppresses DSB formation in the first place versus promotes their rapid repair will require more detailed biochemical studies. These functions may not be mutually exclusive. For example, it is possible that Chd1 facilitates the religation activity of Top2a, and in parallel recruits DNA repair factors as a back-up mechanism that may be particularly important in hypertranscribing cells. In either scenario, Chd1 activity at transcribed promoters protects genome integrity, a finding with significant implications in stem cell biology and cancer.”

Reviewer #3 (Remarks to the Author):

The authors have considerably improved the manuscript and addressed all comments satisfactorily. It is now acceptable for publication.

We are grateful to the Reviewer for the comments provided throughout the revision process.

References:

1. Guzman-Ayala, M. *et al.* Chd1 is essential for the high transcriptional output and rapid growth of the mouse epiblast. - PubMed - NCBI. *Development* **142**, 118–127 (2014).
2. Bulut-Karslioglu, A. *et al.* The Transcriptionally Permissive Chromatin State of Embryonic Stem Cells Is Acutely Tuned to Translational Output. *Cell Stem Cell* **22**, 369-383.e8 (2018).
3. Schwanhäusser, B. *et al.* Global quantification of mammalian gene expression control. *Nature* **473**, 337–342 (2011).
4. Cai, B. *et al.* Triptolide impairs genome integrity by directly blocking the enzymatic activity of DNA-PKcs in human cells. *Biomed Pharmacother* **129**, 110427 (2020).
5. Zhang, Z. *et al.* Triptolide interferes with XRCC1/PARP1-mediated DNA repair and confers sensitization of triple-negative breast cancer cells to cisplatin. *Biomed Pharmacother* **109**, 1541–1546 (2019).
6. Farnung, L., Ochmann, M., Engeholm, M. & Cramer, P. Structural basis of nucleosome transcription mediated by Chd1 and FACT. *Biorxiv* 2020.11.30.403857 (2020) doi:10.1101/2020.11.30.403857.

7. Baranello, L. *et al.* DNA break mapping reveals topoisomerase II activity genome-wide. *International journal of molecular sciences* **15**, 13111–13122 (2014).

REVIEWERS' COMMENTS

Reviewer #1 (Remarks to the Author):

The authors have addressed my previous concerns.

Reviewer #2 (Remarks to the Author):

The authors adequately addressed my concerns. Although I still have a concern about the result of the IP experiment, they will be judged by readers.